# Central neural circuitry mediating courtship song perception in male *Drosophila*

Chuan Zhou[1]*, Romain Franconville[1], Alexander G Vaughan[2], Carmen C Robinett[1], Vivek Jayaraman[1], Bruce S Baker[1]*

[1]Janelia Research Campus, Howard Hughes Medical Institute, Ashburn, United States; [2]Cold Spring Harbor Laboratory, New York, United States

**Abstract** Animals use acoustic signals across a variety of social behaviors, particularly courtship. In *Drosophila*, song is detected by antennal mechanosensory neurons and further processed by second-order aPN1/aLN(al) neurons. However, little is known about the central pathways mediating courtship hearing. In this study, we identified a male-specific pathway for courtship hearing via third-order ventrolateral protocerebrum Projection Neuron 1 (vPN1) neurons and fourth-order pC1 neurons. Genetic inactivation of vPN1 or pC1 disrupts song-induced male-chaining behavior. Calcium imaging reveals that vPN1 responds preferentially to pulse song with long inter-pulse intervals (IPIs), while pC1 responses to pulse song closely match the behavioral chaining responses at different IPIs. Moreover, genetic activation of either vPN1 or pC1 induced courtship chaining, mimicking the behavioral response to song. These results outline the aPN1-vPN1-pC1 pathway as a labeled line for the processing and transformation of courtship song in males.

## Introduction

Sexual behaviors normally involve the exchange of species-specific social cues, including chemical, visual, and auditory information, that elicit a set of sex-specific behaviors (*Dulac and Kimchi, 2007*; *Manoli et al., 2013*). Here, we are particularly focused on auditory communication, which requires the production and perception of sound signals that are diversified between species and are utilized by both invertebrates and vertebrates to attract female mates during courtship (*Murthy, 2010*; *Brainard and Doupe, 2013*). *Drosophila melanogaster* males sing a multi-part, species-specific courtship song that is both critical to female receptivity and encourages courtship by nearby males. Although there has been progress in elucidating the circuitry by which these signals are perceived, processed, and integrated with other sensory cues to elicit stereotyped courtship behaviors, there are very significant gaps in our knowledge.

The neural circuits underlying sexually dimorphic behavior in *Drosophila* are specified by the action of the two terminal genes in the sex determination hierarchy, *fruitless (fruM)* and *doublesex (dsx)*. *fruM* and *dsx* act together to specify sexual behaviors by altering the number, morphology, and physiology of a small subset of CNS (Central Nervous System) neurons (*Manoli et al., 2013*; *Pavlou and Goodwin, 2013*; *Yamamoto and Koganezawa, 2013*). *dsx* is expressed in a limited set of neurons within the CNS, as well as in subsets of non-neuronal cells throughout the body (*Rideout et al., 2010*; *Robinett et al., 2010*). In contrast, *fruM* is expressed only in neurons, including ~2% of neurons within the CNS, as well as sensory, motor, and higher-order interneurons, all of which might comprise neural circuits dedicated to sexual behaviors (*Manoli et al., 2005*; *Stockinger et al., 2005*). This has motivated the proposal that *fruM*+ neurons might provide a set of genetically labeled pathways that channel courtship-related sensory information through a dedicated circuit (*Manoli et al., 2005*; *Stockinger et al., 2005*). Supporting this view, a *fruM*-labeled pathway has been identified that links olfactory reception of a male-specific pheromone to descending neurons that

*For correspondence:
fruitflyroom@gmail.com (CZ);
bakerb@janelia.hhmi.org (BSB)

Competing interests: The authors declare that no competing interests exist.

**eLife digest** The seemingly simple fruit fly engages in an intricate courtship ritual before it mates. Male flies use their wings to 'sing' a complex song that makes females more willing to mate. The song also encourages nearby males to start courting, and these males may then intervene to compete for the female. Each species of fruit fly has its own song, and it is important for both males and females to detect the right song.

The sounds of the courtship song are detected by vibration-sensitive neurons on the flies' antennae. These neurons send signals to the fly's brain. But little is known about how this information is then processed, or how information about the song can be integrated with other courtship cues.

Zhou et al. have now identified a pathway of neurons in male flies that is responsible for hearing the courtship song. This pathway stretches from the antennae to neurons deep within the brain. These neural pathways are different in males and females, suggesting that the two sexes use different circuits of neurons for hearing courtship songs. Zhou et al. then used genetic techniques to show that males need every neuron in this pathway to hear courtship songs.

Further experiments revealed that stimulating the 'deep layer' neurons caused male flies to respond as if they are hearing the courtship song. These neurons are likely to integrate the song with information from other senses and may encode a general signal for arousal.

These findings now pave the way to deepen our understanding of how information from different senses—for example, courtship songs, visual cues, and pheromones—can be integrated to drive specific behaviors. The next challenge is to explore how species-specific songs are detected and recognized, a goal that has yet to be achieved in any species.

drive motor output (*Kurtovic et al., 2007*; *Ruta et al., 2010*; *Kohl et al., 2013*). Despite tantalizing clues, it remains unknown whether *Drosophila* audition is mediated by such dedicated circuits.

When courting a female, the male vibrates his wing to generate a courtship song consisting of two parts: a rhythmic pulse song with an ~35-ms inter-pulse interval (IPI) that strongly elicits male courtship behavior as well as female receptivity, as well as a ~160-Hz sine song that may play a secondary role (*Shorey, 1962*; *Bennet-Clark and Ewing, 1967*; *von Philipsborn et al., 2011*; *Arthur et al., 2013*). *Drosophila* species exhibit significant diversity in the IPI of pulse song, and in *D. melanogaster*, IPIs around ~35 ms are critical for enhancing the responses to courtship song of both conspecific males and females (*Ewing and Bennet-Clark, 1968*; *Bennet-Clark, 1969*; *Cowling and Burnet, 1981*; *Yoon et al., 2013*). In females, pulse song induces reduced locomotion (allowing courting males to come closer) and increases their receptivity to mating attempts (*Bennet-Clark, 1969*; *Schilcher, 1976a*, *1976b*; *Rybak et al., 2002*; *Shirangi et al., 2013*). In males, pulse song elicits both increased locomotion and exploratory courtship activity directed towards nearby flies (*Schilcher, 1976b*; *Eberl et al., 1997*; *Kowalski et al., 2004*; *Kamikouchi et al., 2009*; *Vaughan et al., 2014*). Although males and females show very different motor output in response to song, significant sexual dimorphisms in the sensory pathway have not been identified.

*Drosophila* receives sound stimuli by vibration of the arista, a feather-like structure protruding from the second segment of the antenna (*Gopfert and Robert, 2001*). Vibration of the arista activates the mechanosensory neurons of the Johnston's organ of the second antennal segment, which project to the antennal mechanosensory and motor center (AMMC) of the central brain where they connect with secondary auditory projection neurons (aPNs), local neurons (aLNs), and the giant fiber neurons (*Gopfert et al., 2006*; *Kamikouchi et al., 2009*; *Yorozu et al., 2009*; *Effertz et al., 2011*, *2012*; *Tootoonian et al., 2012*; *Lehnert et al., 2013*; *Pezier et al., 2014*). Among a variety of aPNs, only the aPN1 cell type projecting to the wedge of the ventrolateral protocerebrum (WED) is necessary for song responses in either sex (*Kamikouchi et al., 2009*; *Lai et al., 2012*; *Tootoonian et al., 2012*; *Vaughan et al., 2014*). aPN1 is necessary for both female receptivity and the male song-induced locomotion response, and it shows a response to courtship song that is proportional to pulse rate (*Vaughan et al., 2014*). By identifying downstream neurons in this pathway, we hope to identify the sexually dimorphic structure of this critical pathway, as well as elucidate how song is transformed before reaching the central drivers of courtship output.

One candidate neuronal cluster for central integration of courtship song and other courtship modalities is the set of *dsx*+ pC1 neurons that innervate the lateral protocerebral complex (LPC), a region of dense innervation by both *fru^M*+ and *dsx*+ neurons (*Cachero et al., 2010*; *Rideout et al., 2010*; *Robinett et al., 2010*; *Yu et al., 2010*). These neurons are activated by courtship stimuli in both sexes, but they drive differential behavioral outputs in each sex. Female pC1 neurons are activated by the male-specific pheromone cVA (*Zhou et al., 2014*), while male *fru^M*+/*dsx*+ P1 neurons (a subset of the *dsx*+ pC1 population) are inhibited by cVA but activated by female pheromones (*Kohatsu et al., 2011*). Female pC1 neurons are also activated by sine and pulse songs (*Zhou et al., 2014*). These observations are consistent with pC1 neurons as a site for multimodal integration of courtship stimuli, but how song information is relayed from aPN1 neurons to the pC1/P1 neurons is unknown.

Here, we used a large-scale intersectional screen to identify a labeled line of *fru*+ neurons that supports courtship hearing in male flies. This approach identified a critical *fru^M*+ interneuron type, which we designate as ventrolateral protocerebrum Projection Neuron 1 (vPN1), whose neurites lie in close proximity to those of aPN1 in the WED area. We present several lines of evidence suggesting that vPN1 may represent the third-order auditory neurons. First, anatomical registration of aPN1 and vPN1 suggests axon/dendrite overlap in the WED. Second, genetic inactivation of vPN1 recapitulates the attenuation of male song responses observed for aPN1, while optogenetic activation of vPN1 mimicked pulse song stimuli to induce robust male chaining. Third, GCaMP imaging revealed that both pulse song and sine song elicit strong calcium responses in vPN1 cell bodies and show a preferential response to pulse songs with long IPIs.

In addition, we found that vPN1 neurons likely target *dsx*+ pC1 neurons directly. Anatomically, vPN1 neurons overlap with pC1 neurons in the LPC region. Physiologically, the tuning curve of the pC1 calcium responses to pulse song IPIs closely matches that of behavioral chaining responses. Behaviorally, silencing pC1 neurons reduced male-chaining behavior in response to pulse song, while pC1 activation is sufficient to induce robust chaining responses. Lastly, simultaneous GCaMP imaging and CsChrimson stimulation reveals that the aPN1-vPN1-pC1 pathway is indeed functionally connected. Taken together, we provide anatomical, behavioral, and physiological evidence that the aPN1-vPN1-pC1 pathway provides a labeled line for processing courtship song in *Drosophila*.

## Results

### An intersectional screen identifies *fru^M*+ second- and third-order auditory neurons

Motivated by the hypothesis that *fru^M* labels neurons that detect courtship-relevant sensory stimuli (*Manoli et al., 2005*; *Stockinger et al., 2005*), we performed an anatomical screen aimed at identifying *fru^M*+ neurons in the auditory pathway. Specifically, ~1000 *cis* regulatory module (CRM) GAL4 lines with relatively sparse neuronal expression patterns (*Jenett et al., 2012*) were crossed to *LexAop2-FLP*; *fru^LexA, UAS>stop>myr::GFP* to restrict expression of GFP to those neurons that express both GAL4 and *fru^LexA* (*Figure 1J*). These intersectional expression patterns were then registered onto a standard brain for analysis of potentially overlapping projection patterns.

To characterize *fru^M*+ aPN1 cells, we identified GAL4 drivers (*R21B12*, *R22B11*, and *R49F09*) that label four to five aPN1 cells per hemisphere when intersected with *fru^LexA* (*Figure 1A*, *Figure 1—source data 1*). These intersectional drivers appear to label the same cell type (*Figure 1A*), as co-registered expression patterns are morphologically indistinguishable and consistent with the aPN1/AMMC-B1 cell type (*Kamikouchi et al., 2009*; *Lai et al., 2012*; *Vaughan et al., 2014*).

In order to identify putative third-order auditory neurons, we focused on drivers labeling *fru^M*+ neurons that innervate the WED and project to the central brain. Two GAL4 drivers (*R72E10* and *R46F09*) were identified that, when crossed to *LexAop2-FLP*; *fru^LexA, UAS>stop>myr::GFP*, labeled a subset of the *fru^M*+ aSP-k cell type previously identified via MARCM (Mosaic analysis with a repressible cell marker) clones (*Cachero et al., 2010*) (*Figure 1B* and *Figure 1—source data 1*). These neurons, termed vPN1, have cell bodies located laterally in the dorsal anterior brain and neurites that innervate the WED and the LPC (*Figure 1B*). Co-registration of *fru^M*+ aPN1 and vPN1 neurons revealed substantial overlap in the WED region (*Figure 1C*), suggesting potential synaptic connectivity between aPN1 and vPN1 neurons.

The intersectional labeling of aPN1 and vPN1 was dependent on *fru^LexA* expression. Since many *fru^M*+ neurons have been shown to be sexually dimorphic at the anatomical level, we examined

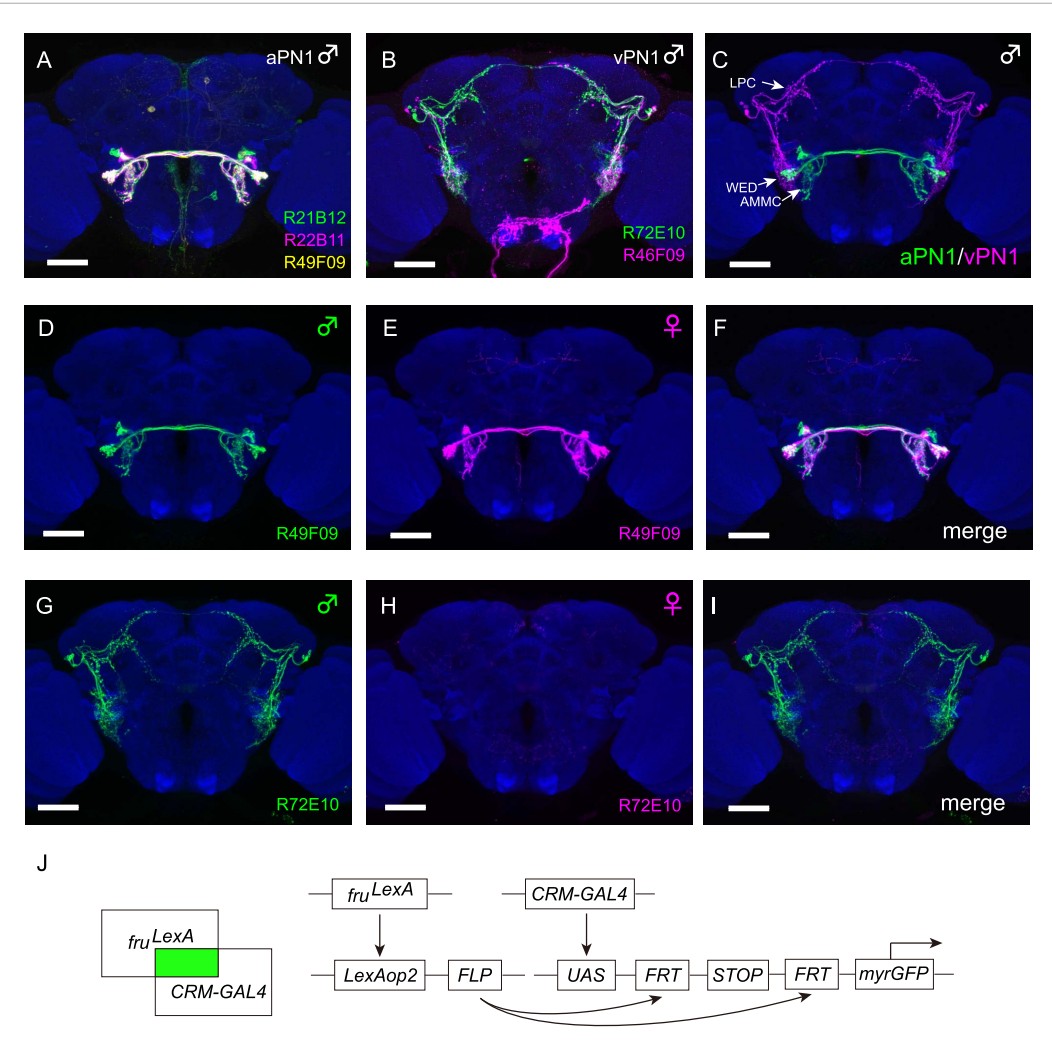

**Figure 1**. Intersectional labeling of auditory neurons. (**A**) Intersectional labeling of male aPN1 neurons using three independent GAL4 lines shown registered onto a standard brain (*R21B12-GAL4* in green, *R22B11-GAL4* in magenta, and *R49F09-GAL4* in yellow). White region is a result of overlapping between different color channels. (**B**) Intersectional labeling of male ventrolateral protocerebrum Projection Neuron 1 (vPN1) neurons using *R72E10-GAL4* (green) and *R46F09-GAL4* (magenta). (**C**) Co-registration of male aPN1 neurons (green) and vPN1 neurons (magenta) onto the standard brain shows significant overlap in WED. aPN1 and vPN1 neurons are labeled by *R49F09-GAL4* ∩ *fru$^{LexA}$* and *R72E10-GAL4* ∩ *fru$^{LexA}$*, respectively. (**D–F**) aPN1 neurons are present in both male (**D**) and female (**E**) brains of *LexAop2-FLP/+; fru$^{LexA}$, UAS>stop>myr::GFP/49F09-GAL4* flies. (**F**) Merge of (**D**) and (**E**). (**G–I**) vPN1 neurons are present in male (**G**) but absent in female (**H**) brains of *LexAop2-FLP/+; fru$^{LexA}$, UAS>stop>myr::GFP/72E10-GAL4* flies. (**I**) Merge of (**G**) and (**H**). (**J**) Schematic drawing of GAL4/LexA intersection for labeling subsets of *fru$^M$+* neurons. FLP expression induced by *fru$^{LexA}$* will remove the stop cassette to allow GFP reporter expression only in neurons expressing both *fru$^{LexA}$* and the *CRM-GAL4*. Scale bars, 50 μm. All images were aligned and registered onto a standard brain.

The following source data is available for figure 1:

**Source data 1**. Quantification of aPN1 or VPN1 neurons labeled by intersectional drivers.

whether the morphologies of either the aPN1 or vPN1 neurons were sexually dimorphic. aPN1 neurons had indistinguishable projection patterns in female and male brains (*Figure 1D–F*), while vPN1 neurons were only observed in male brains (*Figure 1G*), with female brains showing only very weak expression in a few seemingly unrelated neurons (*Figure 1H–I*). However, aSP-k neurons are present in both sexes and extend male-specific processes that innervate the LPC arch (*Cachero et al., 2010*).

In addition, we identified a split-GAL4 combination (*R72E10-GAL4AD ∩ VT9665-GAL4DBD*, referred to as vPN1 split-GAL4 hereafter) that labels vPN1 neurons in the male brain (*Figure 2A*), independent of *fru^LexA* expression. Using this driver, vPN1 neurons remain absent from the female brain, and *fru^M* is expressed in all vPN1 cells of the male (*Figure 2B,C*). We therefore infer that the vPN1 population constitutes a male-specific subset of the larger aSP-k cell type.

We next asked whether *fru^M* is necessary and sufficient for specifying the male-specific vPN1 neurons. We did this by examining vPN1 split-GAL4 expression patterns in various *fru* mutant backgrounds. The male-specific expression pattern of vPN1 is not affected in heterozygous *fru* mutants (*fru^{4–40}/+*, *Figure 2D*). However, vPN1 expression is absent in *fru* null mutant males (*fru^{4–40}/fru^LexA*, *Figure 2E*), while misexpression of Fru^M protein in females (*fru^M/+*) leads to an induction of vPN1 expression in the female brain (*Figure 2F*) (*Demir and Dickson, 2005*). Thus, *fru^M* plays an instructive role in the specification of sexual dimorphism of vPN1 neurons.

Our anatomical results suggest a candidate pathway that may carry auditory information from aPN1 to higher brain regions via male-specific vPN1 projections. We next tested whether these neurons are necessary for male courtship hearing.

## Song-induced chaining behavior is tuned to conspecific IPIs

Male flies respond to courtship song, and pulse song in particular, by increasing locomotion and exploratory courtship of nearby flies (*Schilcher, 1976b*; *Eberl et al., 1997*; *Kowalski et al., 2004*). Ecologically, this response is appropriate for the context of competitive courtship on fruit substrates. Experimentally, stimulation of groups of males with synthetic courtship song can induce both locomotion and chains of males engaging in the initial stages of courtship. This response (distinct from the unstimulated and dysregulated courtship observed in *fru* mutant males) has been used to investigate the genetic and neural mechanisms underlying auditory detection (*Eberl et al., 1997*; *Kamikouchi et al., 2009*; *Vaughan et al., 2014*). To assay the role of putative auditory interneurons in male courtship hearing, we constructed courtship chambers with sloped side walls for better visualization of song-induced chaining behavior (*Figure 3A* and *Figure 3—figure supplement 1*) (*Simon and Dickinson, 2010*). Upon stimulation by pulse song with a ramping intensity, we indeed observed that male flies displayed robust song-induced chaining behavior (*Figure 3A*).

*Drosophila* pulse songs have two key features: intra-pulse frequency (IPF) and IPI. Across different *Drosophila* species, the antennal receiver is tuned to the conspecific IPF (*Riabinina et al., 2011*), and behavioral responses appear tuned to the conspecific IPI (*Ewing and Bennet-Clark, 1968*; *Cowling and Burnet, 1981*). While mechanical features of the antenna may have evolved to optimally detect conspecific IPFs, it has been hypothesized that the brain is responsible for recognizing conspecific IPIs (*Riabinina et al., 2011*). Therefore, we asked whether song-induced chaining behavior of *D. melanogaster* males varied as a function of song IPI (*Schilcher, 1976b*; *Yoon et al., 2013*). Indeed, the chaining response was significantly higher for songs at 35-ms IPI than for all other IPIs, across a wide range of intensities ($p < 0.05$, 72.5–85 dB, Wilcoxon rank-sum test; *Figure 3B,C*), suggesting that *D. melanogaster* males to some extent are able to preferentially respond to pulse song of their own species. A similar effect was observed using intermittent song stimuli (80 dB; *Figure 3D,E*). We therefore used this assay to test the behavioral role of aPN1, vPN1, and pC1 populations.

## Silencing second- and third-order neurons decreased song-induced chaining behavior

aPN1 neurons had been identified as putative second-order auditory neurons for courtship hearing, based on their anatomical location, physiological response to song, and the behavioral phenotypes produced in response to courtship song in both male and female with aPN1 neurons silenced (*Vaughan et al., 2014*). We asked whether the *fru^M+* subset of aPN1 is also required for courtship hearing in males. Three independent *GAL4 ∩ fru^LexA* genotypes were used to confirm that silencing *fru^M+* aPN1 neurons reduced male courtship hearing; indeed, flies expressing tetanus neurotoxin light chain (TNT), which blocks synaptic vesicle release, in *fru^M+* aPN1 neurons showed significantly reduced chaining compared to controls expressing an inactive form of TNT (TNT^in) or lacking *fru^LexA* (*Figure 4A–C*) (*Sweeney et al., 1995*). Thus, *fru^M+* aPN1 neurons are required for song-induced male chaining. We similarly asked whether the vPN1 neurons were required for the male-chaining response to song. Using two GAL4 drivers (*R72E10*, *R46F09*) in an intersectional approach (*GAL4 ∩ fru^LexA*) to

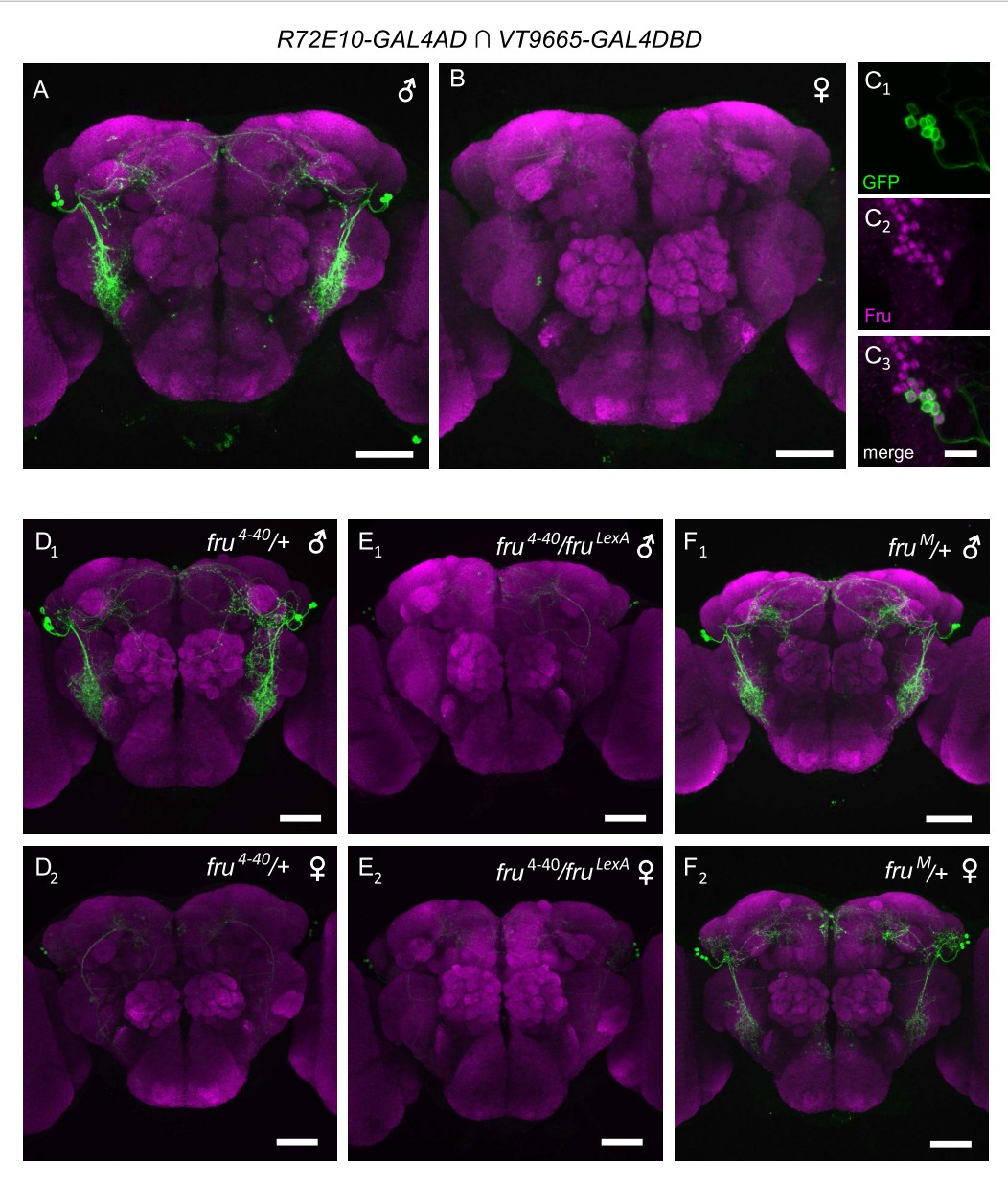

**Figure 2**. *fru* is necessary and sufficient for specifying male-specific vPN1 neurons. (**A**, **B**) GFP expression (green) in the male (**A**) or female (**B**) brain of *R72E10-GAL4AD/UAS-mCD8GFP; VT9665-GAL4DBD/+* flies counter-stained by nc82 antibody (magenta). vPN1 neurons with VLP projections were labeled in the male but not female brain. (**C**) vPN1 neurons (**C₁**) in the male brain of *R72E10-GAL4AD/UAS-mCD8GFP; VT9665-GAL4DBD/+* flies co-stained with Fru$^M$ antibody (**C₂**). (**C₃**) is a merge of (**C₁**) and (**C₂**). (**D**) vPN1 neurons are present in the male brain (**D₁**) but not female brain (**D₂**) of *fru$^{4-40}$/+* flies. Genotype is: *R72E10-GAL4AD/UAS-mCD8GFP; VT9665-GAL4DBD, fru$^{4-40}$/+*. (**E**) vPN1 neurons are absent in both the male brain (**E₁**) and female brain (**E₂**) of null mutant *fru$^{4-40}$/fru$^{LexA}$*. Genotype is: *R72E10-GAL4AD/UAS-mCD8GFP; VT9665-GAL4DBD, fru$^{4-40}$/fru$^{LexA}$*. (**F**) vPN1 neurons are present in both the male brain (**F₁**) and female brain (**F₂**) of *fru$^M$* mutant flies where Fru$^M$ is expressed in both males and females. Genotype is: *R72E10-GAL4AD/UAS-mCD8GFP; VT9665-GAL4DBD, fru$^M$/+*.

target TNT expression to vPN1 neurons, we observed that vPN1 silencing reduced song-induced chaining (*Figure 4D,E*).

As the multiple drivers used for aPN1 and vPN1 share no off-target expression in *fru$^{LexA}$*+ neurons (*Figure 4A–E*), we conclude that both *fru$^M$*+ aPN1 and *fru$^M$*+ vPN1 *fru$^M$* neurons are required for the male-chaining response to pulse song.

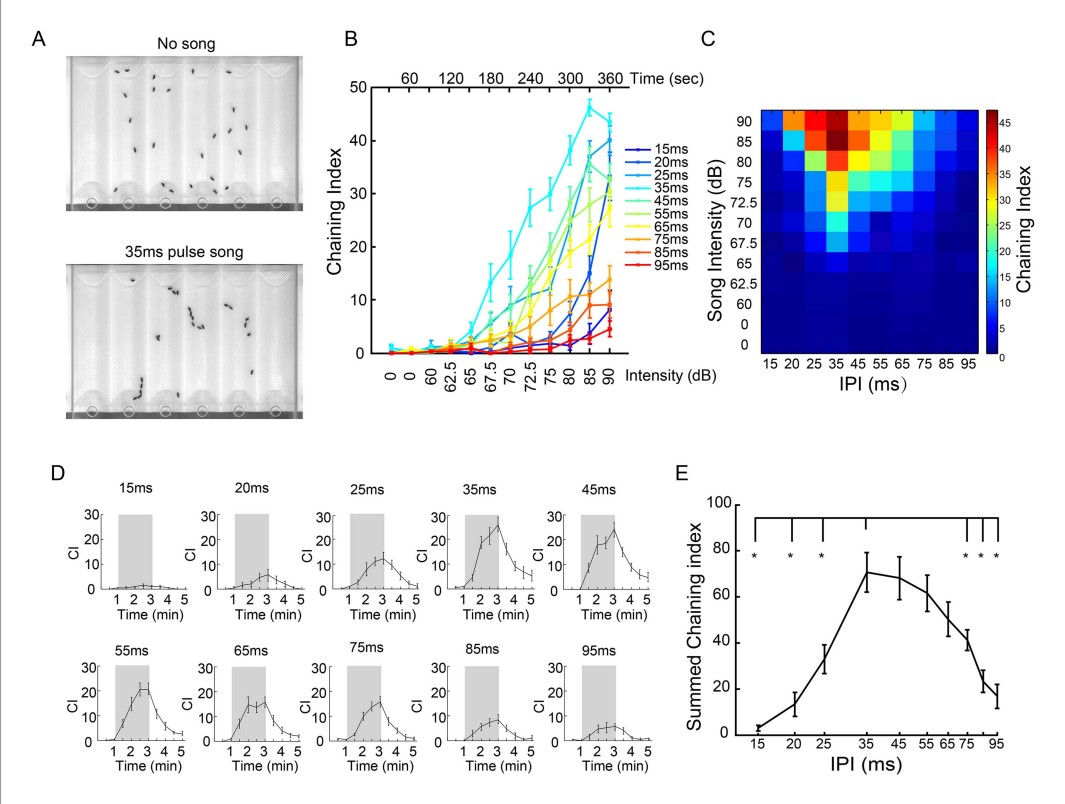

**Figure 3**. IPI tuning of song-induced male-chaining behavior. (**A**) Song-induced chaining assay. Males court each other when exposed to pulse song with a 35-ms inter-pulse interval (IPI) and robustly form courtship chains (bottom). (**B**) Chaining indices (CIs) of wild-type males in response to different IPIs. After 60 s of silence, continuous pulse songs were played back with a ramping sound intensity from 60 dB to 90 dB. The intensity was increased every 30 s. n = 14 for all groups. Error bars represent SEM. (**C**) Heat map visualization of chaining responses in (**B**). Each cell represents the chaining index at a given IPI and song intensity. (**D**) CIs of wild-type males in response to intermittent pulse songs with different IPIs at 80 dB. A train of 40 pulses was delivered every five seconds so that the number of pulses was the same for different IPIs. After one minute of silence, two minutes of intermittent pulse song stimuli were presented (indicated by shadowed box), and then followed by two minutes of silence. Chaining persisted and gradually decreased after song presentation. n = 15 for all groups. (**E**) CIs during the song presentation period in (**D**) were summed up and plotted as a function of IPIs. Error bars represent SEM. *p < 0.05 when chaining responses at 35-ms IPI were compared to those at 15-, 20-, 25-, 75-, 85-, and 95-ms IPIs (Wilcoxon rank-sum test).

The following figure supplement is available for figure 3:

**Figure supplement 1**. Song-induced chaining setup.

## Auditory responses of vPN1 neurons

Courtship song responses in aPN1 have been extensively studied using in vivo calcium imaging and electrophysiology (*Lai et al., 2012*; *Tootoonian et al., 2012*; *Vaughan et al., 2014*). To investigate the physiological function of downstream vPN1 neurons in auditory perception, we performed in vivo calcium imaging in vPN1 cell bodies during song presentation by expressing *UAS-GCaMP6m* under the control of *R72E10-GAL4* (*Figure 5A–C*) (*Chen et al., 2013*).

When auditory stimuli were presented at 80 dB, vPN1 neurons responded to both pulse song and sine song, but not to white noise stimuli (*Figure 5D,E*). We then looked at the response of vPN1 neurons to pulse songs of different intensities. Strikingly, the threshold of vPN1 activation matched the behavioral threshold for song-induced chaining (EC50 = 68.2 dB for vPN1 responses v.s. EC50 = 70.6 dB for chaining responses, half-maximal sigmoid fit; *Figure 5F* and *Figure 3B*).

To further understand how vPN1 neurons process pulse song, we examined their tuning with respect to a variety of IPIs from 15 ms to 95 ms at 80 dB (*Figure 5D,G*). We found that vPN1 neurons

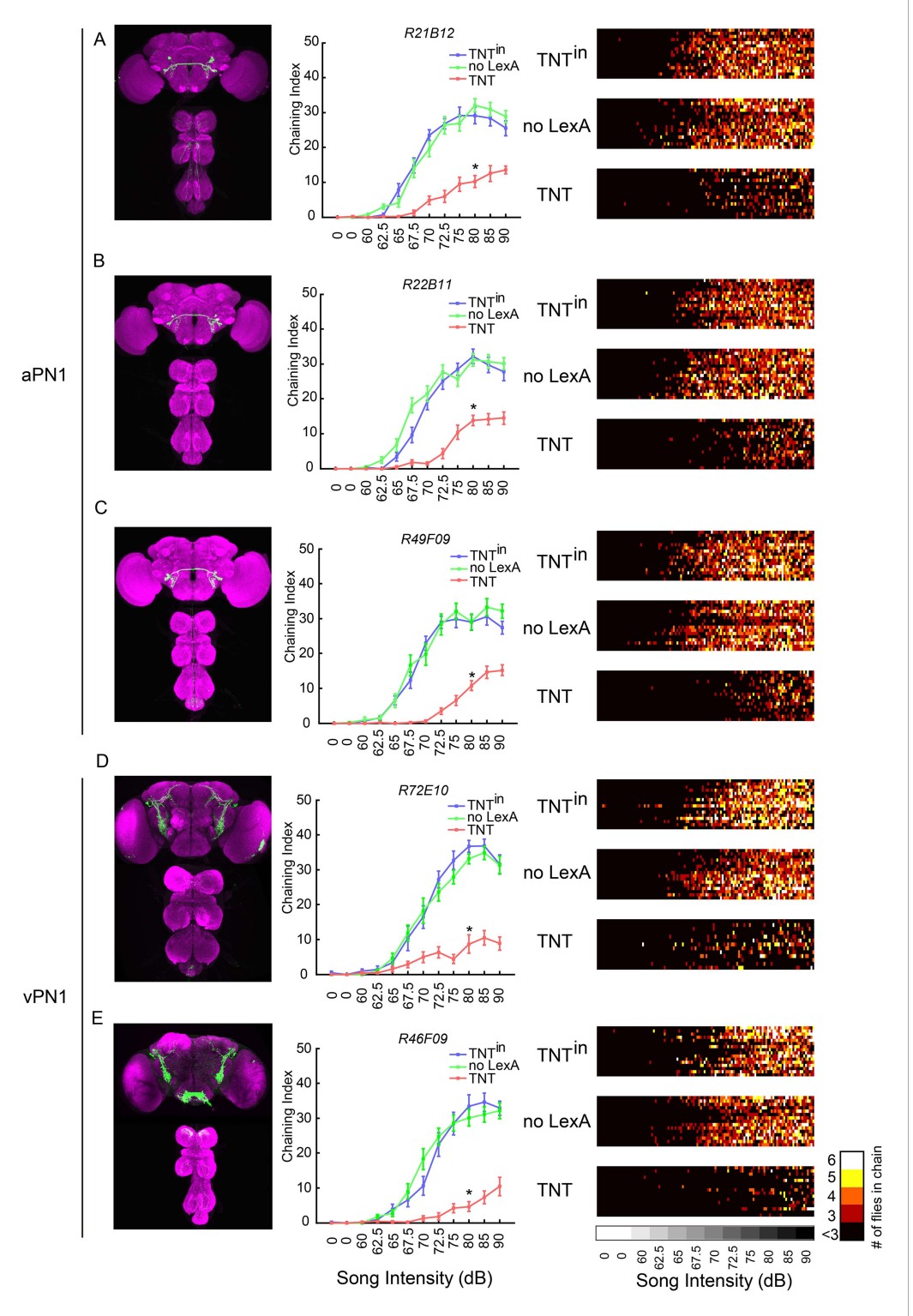

**Figure 4**. Inactivation of second- and third-order auditory neurons reduced chaining responses to pulse song. (**A–C**) Silencing aPN1 neurons decreased song-induced chaining responses. aPN1 drivers *R21B12-GAL4* (**A**), *R22B11-GAL4* (**B**), or *R49F09-GAL4* (**C**) were crossed to *UAS>stop>TNT; fru^{LexA}, LexAop2-FLP* (TNT group), *UAS>stop>TNT^{in}; fru^{LexA}, LexAop2-FLP* (TNT^{in} group), *or UAS>stop>TNT; LexAop2-FLP* (no LexA group). (**D**, **E**) Silencing vPN1 neurons decreased song-induced chaining responses. vPN1 drivers *R72E10-GAL4* (**D**), *R46F09-GAL4* (**E**) were crossed to *UAS>stop>TNT; fru^{LexA}, LexAop2-FLP* (TNT group), *UAS>stop>TNT^{in}; fru^{LexA},*

*Figure 4. Continued*

*LexAop2-FLP* (TNT[in] group), *or UAS>stop>TNT; LexAop2-FLP* (no LexA group). n = 13–16 for each condition. *p < 0.0001 compared to both controls at 80 dB, Wilcoxon rank-sum test. Shown in the left panel is GFP expression of each intersectional driver. In the right panel, a heat map summary shows chaining intensities across the testing time course. Each row corresponds to a group of six flies. Colors represent the number of flies in chain.

respond strongly to pulse trains with IPIs of 35 ms and above, with significant attenuation at IPIs below 25 ms (p < 0.01 when compared to responses at 35-ms IPI, Wilcoxon signed-rank test; *Figure 5H* and *Figure 5—figure supplement 1*). This signal displays low-pass response properties for IPI with a shoulder around 35-ms and is different from the aPN1 response that responds as a function of pulse rate (*Vaughan et al., 2014*).

## The role of pC1 *dsx* neurons in song perception

The vPN1 neurons extend projections into the LPC, an area surrounding the mushroom body peduncle that is enlarged in males and densely innervated by both *fru[M]*+ and *dsx*+ neurons (*Figure 1B*) (*Cachero et al., 2010*; *Rideout et al., 2010*; *Robinett et al., 2010*; *Yu et al., 2010*). One of the cell types innervating this region is the male-specific *fru[M]*+/*dsx*+ P1 cell type (a subset of pC1), which controls the initiation of male courtship (*Kimura et al., 2008*; *Kohatsu et al., 2011*; *Pan et al., 2011*; *von Philipsborn et al., 2011*). Some pC1 neurons are present in both sexes, although their projections and cell number are highly sexually dimorphic (*Rideout et al., 2010*; *Robinett et al., 2010*; *Zhou et al., 2014*). Female pC1 neurons are sensitive to courtship song and the male-specific pheromone cVA, suggesting a role for female pC1 neurons in integrating multiple courtship-related sensory signals (*Zhou et al., 2014*).

We asked whether male pC1 neurons function downstream of vPN1 neurons to process song signals. To evaluate the potential connections between pC1 neurons and vPN1 neurons, we first labeled pC1 neurons via the intersection of *R71G01-LexA::p65* with *dsx[GAL4]* (*Pan et al., 2012*) and then registered male pC1 neurons and vPN1 neurons onto a standard brain (*Figure 6A*). The neurites of pC1 and vPN1 neurons extensively overlap in the LPC, including the lateral crescent, the lateral junction, the arch and the ring region (*Yu et al., 2010*), suggesting potential synaptic contacts between these two cell types (*Figure 6B–D*). To examine the roles of pC1 neurons in song perception, *R71G01-LexA::p65* ∩ *dsx[GAL4]* was used to express TNT in *dsx*+ pC1 neurons. Compared to controls, TNT-mediated inactivation of pC1 neurons almost completely abolished song-induced chaining behavior (*Figure 6E*).

To ask whether male pC1 neurons also respond to song input, we expressed GCaMP6m in *dsx[GAL4]* neurons and recorded the activity of the pC1 neurites in the LPC while presenting song stimuli (*Figure 6F*). pC1 neurons are only sensitive to pulse song stimuli at 80 dB and above (EC50 = 79.7 dB, sigmoid fit; *Figure 6I*), a higher activation threshold than that observed for vPN1 (EC50 = 68.2 dB; *Figure 5F*). This could be due to intrinsic properties of pC1 neurons, which might require multi-sensory input to become fully activated. As expected, pulse-song stimuli evoked calcium transients in pC1 neurites, while sine song and white noise induced little response (*Figure 6H*), consistent with the report that sine song and white noise are not capable of inducing chaining behavior (*Eberl et al., 1997*).

We next examined the IPI tuning of male pC1 neurons. Unlike the low-pass response of vPN1, the pC1 neurite response showed a strong response to pulse song with IPIs of 35–65 ms. This response is significantly reduced at both short (15–25 ms) and long (85–95 ms) IPIs (p < 0.01 when compared to responses at 35-ms IPI, Wilcoxon signed-rank test; *Figure 6K*, *Figure 6—figure supplement 1*). This response reflects a band-pass sensitivity to pulse song, which is not seen in either aPN1 or vPN1, and qualitatively matches the IPI sensitivity of the behavioral response in male and female flies (*Bennet-Clark, 1969*; *Yoon et al., 2013*; *Vaughan et al., 2014*).

## Potential transformation of song responses between vPN1 and pC1

The auditory pathway leading through aPN1, vPN1, and pC1 shows a possible transformation of song representation. In particular, while aPN1 responses integrate pulse rate at IPIs longer than 25 ms (*Vaughan et al., 2014*), we found that vPN1 shows a low-pass response and pC1 shows a band-pass response to IPI (*Figure 6—figure supplement 2A*). We compared vPN1 and pC1 responses and

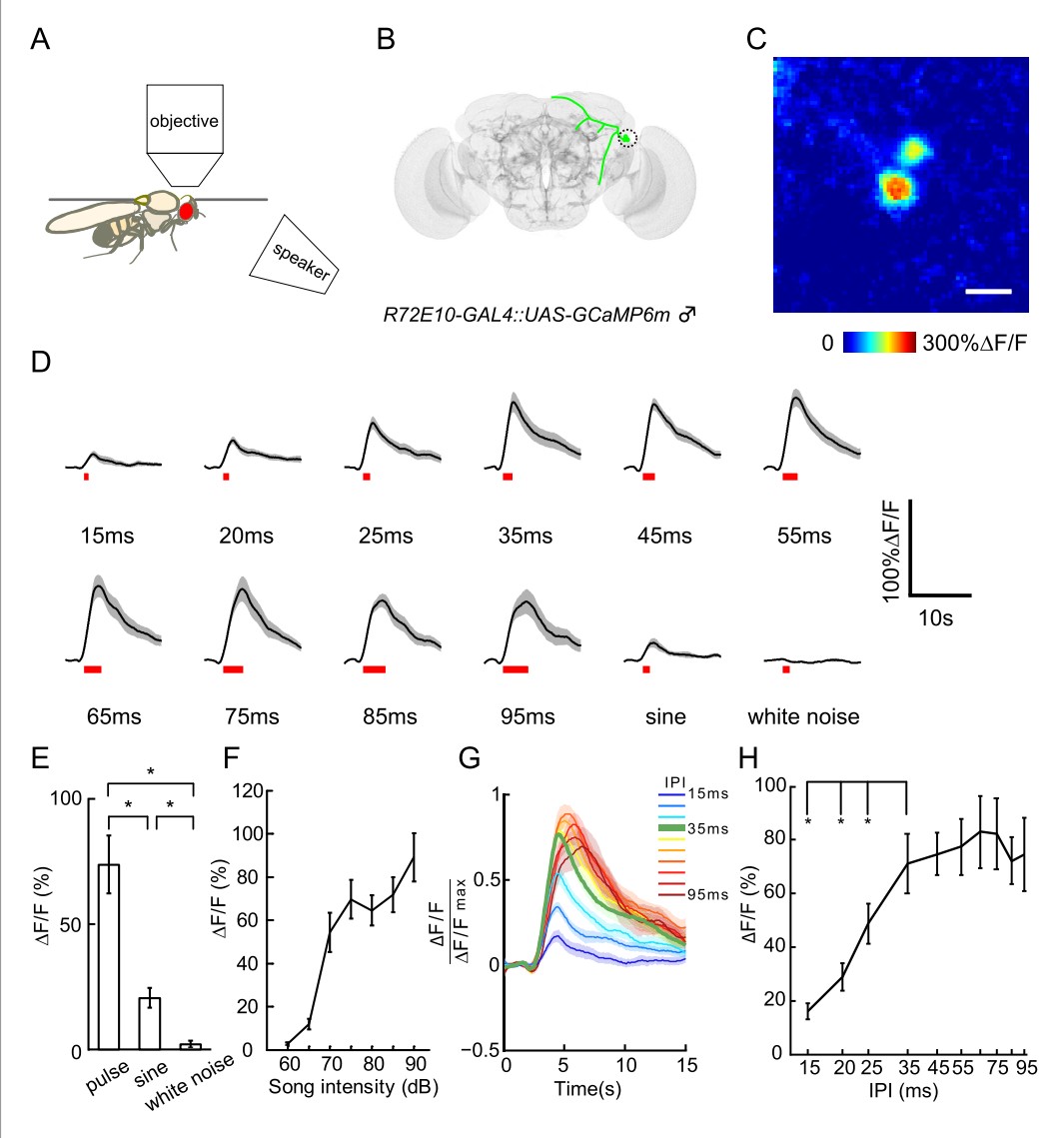

**Figure 5**. Calcium responses of vPN1 neurons to courtship song. (**A**) Diagram of imaging setup in which a speaker was located 20-cm away from the recorded male fly. (**B**) Diagram of vPN1 neurons labeled with *R72E10-GAL4* driving expression of GCaMP6. Cell bodies are circled. (**C**) Calcium imaging of *R72E10-GAL4* driven GCaMP6 expression in vPN1 cell bodies. Heat map of a sample frame shows ΔF/F changes in two vPN1 cell bodies. Scale bar, 10 μm. (**D**) Calcium responses of vPN1 neurons to a train of 40 pulses at different IPIs, sine song, and white noise at 80 dB. Black lines represent means. Gray envelopes indicate SEM. Song stimulus durations are indicated as red bars below. (**E**) Peak ΔF/F changes of vPN1 neurons stimulated with pulse song (35-ms IPI, 40 pulses), sine song (140 Hz, 1.4 s), and white noise (1.4 s) at 80 dB. *p < 0.01, Wilcoxon rank-sum test. n = 10 for all the groups. (**F**) Peak ΔF/F of vPN1 neurons in response to pulse song (35-ms IPI, 40 pulses) at different sound intensities. n = 14 trials for each sound level. (**G**) Normalized calcium traces of vPN1 neurons at different IPIs. Each ΔF/F was normalized by the maximum ΔF/F. (**H**) Peak ΔF/F of vPN1 neurons in response to different IPIs at 80 dB (40 pulses). n = 10 for all groups. *p < 0.01, Wilcoxon signed-rank test.

The following figure supplement is available for figure 5:

**Figure supplement 1**. Raster plots of vPN1 neurons in individual flies.

calculated a direct transfer function for them (***Figure 6N*** and ***Figure 6—figure supplement 2***). This comparison shows significant attenuation of IPIs below (25 ms) and above (75–95 ms) the 35-ms IPI, indicating a possible band-pass filter linking these two responses (Wilcoxon rank-sum test, one-sided).

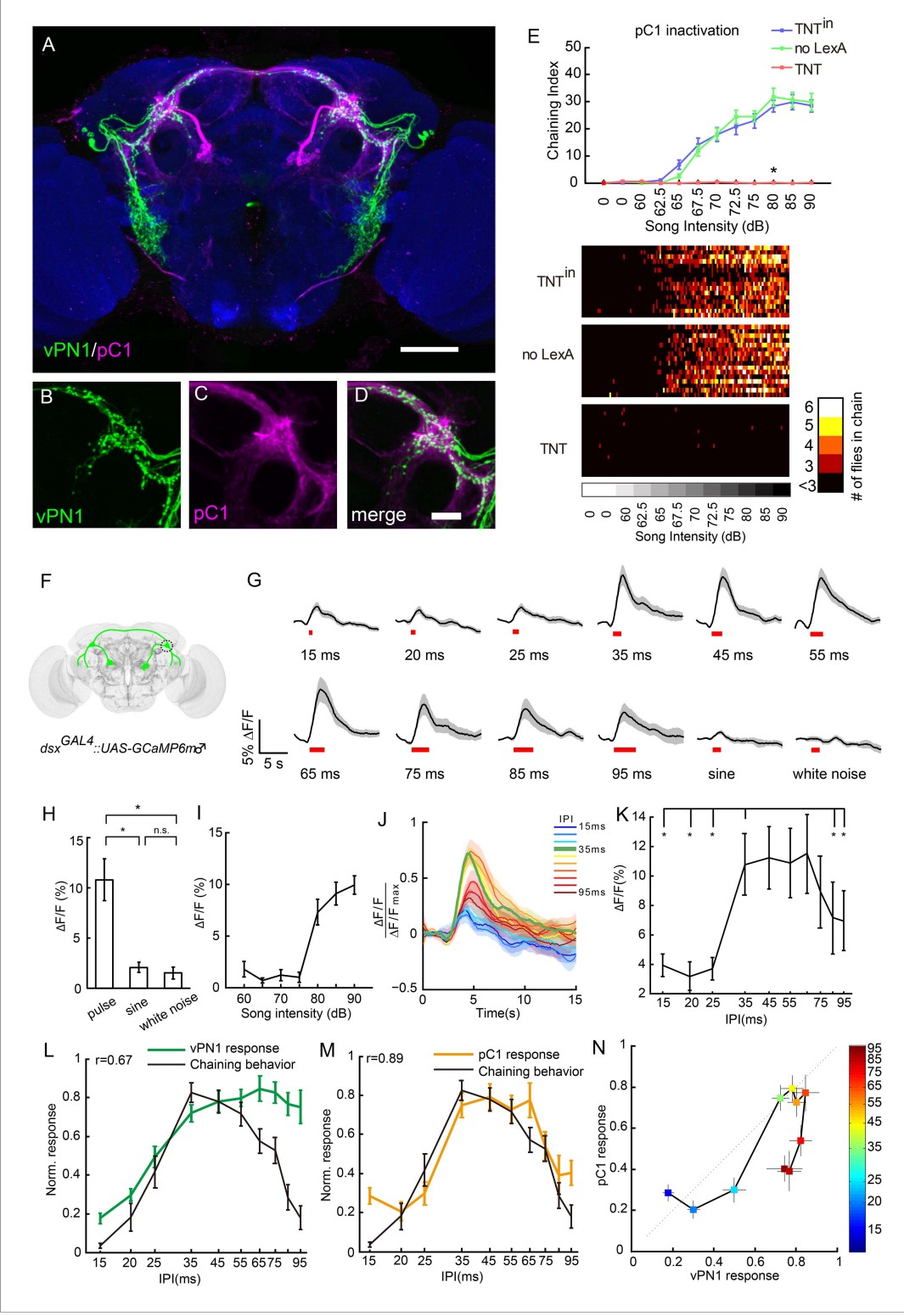

**Figure 6**. Anatomical, behavioral, and physiological characterization of *dsx*+ pC1 neurons in auditory sensation. (**A–D**) Co-registration of vPN1(green) and pC1(magenta) neurons onto the standard brain. Genotype for labeling vPN1 neurons: *LexAop2-FLP/+; fru^LexA, UAS>stop>myr::GFP/R72E10-GAL4*. Genotype for labeling vPN1 neurons: *R71G01-LexA/UAS>stop>myr::GFP; dsx^GAL4, LexAop2-FLP/+*. vPN1 processes (**B**) and pC1 (**C**) processes overlap in the region of lateral protocerebral complex (LPC). (**D**) Merge of (**B**) and (**C**). Scale bars, (**A**) 100 μm and (**D**) 20 μm.
*Figure 6. continued on next page*

*Figure 6. Continued*

(**E**) Song-induced chaining response was impaired by inactivation of pC1 neurons. Genotypes: *UAS>stop>TNT$^{in}$/71G01-LexA; dsx$^{GAL4}$, LexAop2-FLP/+* (TNT$^{in}$), *UAS>stop>TNT/+; dsx$^{GAL4}$, LexAop2-FLP/+* (no LexA), *UAS>stop>TNT/71G01-LexA; dsx$^{GAL4}$, LexAop2-FLP/+* (TNT). n = 16 for all the conditions. *p < 0.0001 when comparing TNT group to both controls at 80 dB, Wilcoxon rank-sum test. Bottom, heat map analysis of chaining events for individual groups of flies. (**F**) Diagram of pC1 neurons labeled with *dsx$^{GAL4}$* driving expression of GCaMP6. Neurites innervating the LPC are circled. (**G**) Calcium responses of pC1 neurons responding to different IPIs, sine song, and white noise at 80 dB. Pulse song stimuli consist of a train of 40 pulses. Black lines indicate mean values, while gray areas indicate SEM. Song stimulus durations are indicated as red bars. (**H**) Peak ΔF/F values of pC1 neurons stimulated with pulse song (35-ms IPI, 40 pulses), sine song (140 Hz, 1.4 s), and white noise (1.4 s) at 80 dB. *p < 0.01, Wilcoxon rank-sum test. No significance was observed for sine vs white noise. n = 12 for all the groups. (**I**) Peak ΔF/F values of pC1 neurons stimulated with pulse song (35-ms IPI, 40 pulses) from 60 dB to 90 dB. pC1 neurons are only sensitive to pulse song stimuli above 80 dB. n = 9 trials for each sound level. (**J**) Normalized calcium traces of pC1 neurons at different IPIs. Each ΔF/F was normalized by the maximum ΔF/F. (**K**) Peak ΔF/F of pC1 neurons stimulated with different IPIs at 80 dB (40 pulses). n = 12 for all groups. *p < 0.01, Wilcoxon signed-rank test. (**L**) Comparison between the calcium response (peak ΔF/F) of vPN1 neurons and chaining responses to different IPIs shown in *Figure 3E* (R = 0.67; p < 0.017, permutation test). Both calcium responses and chaining responses are normalized to their respective maximum responses. (**M**) Comparison between the calcium response (peak ΔF/F) of pC1 neurons and chaining responses to different IPIs shown in *Figure 3E* (R = 0.89; p < 0.001, permutation test). Both calcium responses and chaining responses are normalized to their respective maximum responses. (**N**) Comparison between normalized vPN1 responses and pC1 responses. Each colored square represents a different IPI indicated by the heat map. Error bars represent SEM in all panels.

The following figure supplements are available for figure 6:

**Figure supplement 1**. Raster plots of pC1 neurons in individual flies.

**Figure supplement 2**. Transfer function between vPN1 and pC1 responses to IPIs of the pulse song.

**Figure supplement 3**. Comparison of synthetic courtship song and natural courtship song with a particle-velocity microphone.

We next examined the relationship between the GCaMP response and song-induced chaining behavior more closely. For vPN1, we observed only a moderate correlation between this calcium response and the IPI sensitivity of song-induced chaining (r = 0.67; p < 0.017, permutation test, *Figure 6L*). In contrast, the pC1 response closely matches the IPI sensitivity of song-induced chaining (r = 0.89; p < 0.001, permutation test, *Figure 6M*). The correlation with the behavioral response is higher for pC1 than vPN1 (p < 0.037, Meng's z-test). These results suggest that pC1 activity may integrate information from vPN1 and other neurons to determine the level of courtship-chaining behavior.

As caveats, however, we note that the slow decay kinetics of GCaMP6m (*Chen et al., 2013*) may account for some of the observed low-pass properties observed in our vPN1 or pC1 recordings. In addition, some of the differences between the tuning of vPN1 and pC1 may arise from distinct calcium dynamics in the recording sites for vPN1 (soma) and pC1 (neurites), or from cell-type specific differences in calcium dynamics. Indeed, although our results suggest that song representations may be serially transformed in the ascending auditory pathway, a circuit-level understanding of how local and projection neurons in this pathway shape this response remains to be discovered.

## Optogenetic activation of vPN1 or pC1 neurons induced male chaining

To investigate whether neurogenetic activation of auditory neurons mimics the effects of courtship song presentation in inducing male-chaining behavior, we expressed the red light-sensitive channelrhodopsin CsChrimson in auditory neurons with intersectional drivers and assessed the effects of red light stimulation on groups of male flies (*Klapoetke et al., 2014*).

CsChrimson-mediated activation of aPN1 neurons did not induce male-chaining behavior when we tested two independent GAL4 drivers (*R22B11* and *R49F09*) (*Figure 7A,D*). This is consistent with the report that dTrpA1-mediated hyperactivation of aPN1 neurons failed to restore receptivity of

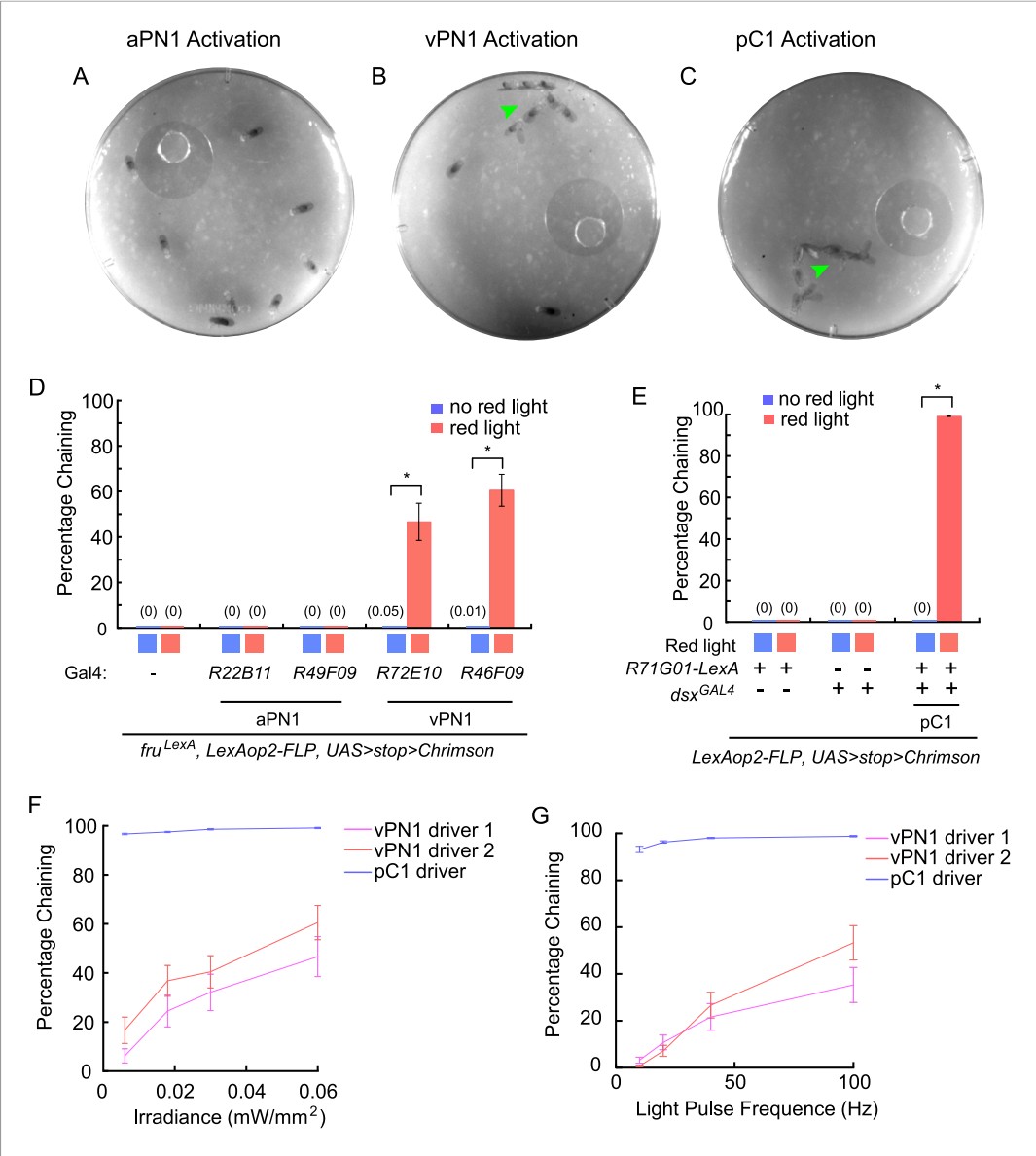

**Figure 7.** Optogenetic activation of auditory neurons. (**A–C**) Optogenetic activation of vPN1 neurons (**B**) or pC1 neurons (**C**) induced male chaining, as evidenced by males courting each other on food (green arrow head), while activation of aPN1 neurons did not (**A**). CsChrimson activation was achieved with constant 655-nm light (0.06 mW/mm$^2$). *R22B11-GAL4 ∩ fru$^{LexA}$, R72E10-GAL4 ∩ fru$^{LexA}$*, or *R71G01-LexA ∩ dsx$^{GAL4}$* was used to drive CsChrimson expression in aPN1 (**A**), vPN1 (**B**), or pC1 (**C**). (**D**) Male-chaining behavior was induced by CsChrimson-mediated activation of vPN1 neurons but not aPN1 neurons. *p < 0.0001 (Student's *t*-test). n = 14–18 for all the genotypes. (**E**) Male-chaining behavior was induced by CsChrimson-mediated activation of pC1 neurons. *p < 0.0001 (Student's *t*-test). n = 16–18 for all the genotypes. (**F, G**) Male-chaining behavior induced by CsChrimson-mediated activation of either vPN1 or pC1 neurons when stimulated with constant red light at 0.006, 0.018, 0.03, 0.06 mW/mm$^2$ (**F**) or stimulated with 5-ms light pulses at 10, 25, 50, and 100 Hz (**G**). vPN1 driver 1 is *R72E10-GAL4 ∩ fru$^{LexA}$*; vPN1 driver 2 is *R46F09-GAL4 ∩ fru$^{LexA}$*, pC1 driver is *R71G01-LexA ∩ dsx$^{GAL4}$*. n = 16–28 for all the genotypes.

females toward wingless males (*Vaughan et al., 2014*). In contrast, CsChrimson-mediated activation of putative third-order vPN1 neurons induced robust chaining behavior (*Figure 7B,D* and *Video 1*), which was not observed in the absence of red light stimulation or in control flies lacking GAL4 (*Figure 7D*). Similarly, activation of pC1 neurons induced robust male-chaining behavior (*Figure 7C,E* and *Video 2*).

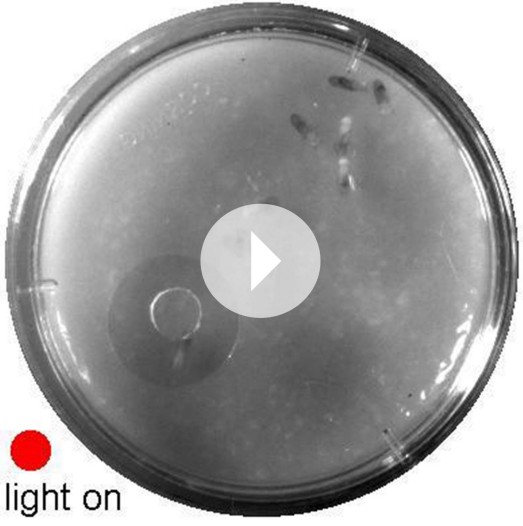

**Video 1.** CsChrimson activation of vPN1 neurons induced chaining behavior in LexAop2-FLP/+; UAS>dsFRT>CsChrimson-mVenus, fruLexA/72E10-GAL4 male flies.

**Video 2.** CsChrimson activation of pC1 neurons induced chaining behavior in LexAop2-FLP/71G01-LexA; UAS>dsFRT>CsChrimson-mVenus/dsxGAL4 male flies.

To probe the sensitivity of the chaining response to red light stimulation, we examined chaining behavior when activating either vPN1 or pC1 with various intensities or frequencies of light (*Figure 7F,G*). While activation of vPN1 induced chaining in an intensity-dependent or frequency-dependent manner, activation of pC1 generated a saturating response in which even the lowest intensity (0.006 mW/mm²) or the lowest frequency (10 Hz) was able to induce robust chaining through >90% of the testing period. Although it is not feasible to control for the expression strength of the aPN1, vPN1, and pC1 drivers, we suspect that the different dynamics of chaining behavior induced by activating aPN1, vPN1, or pC1 neurons may reflect the intrinsic properties of the cell types and the manner in which courtship song is encoded in these neurons. In particular, we note that the lack of response for aPN1 is consistent with the observation that the stimulus that best activates aPN1 (25 ms IPI) does not induce a strong behavioral response from wild-type flies (*Vaughan et al., 2014*).

As song information flows from peripheral to central brain, the higher-order neurons may represent the socially relevant features of courtship in a simpler manner than in lower neurons, and this may account for the gradient of behavior phenotype we observed when we activate aPN1, vPN1, or pC1, respectively. This (still theoretical) architecture describes a possible transformation of song representations from a temporal code in aPN1 to a rate code in vPN1—which is ultimately coupled with other sensory signals to encode overall behavioral arousal in PC1 (*Figure 8F* and *Video 3*).

## The aPN1-vPN1-pC1 pathway is functionally interconnected

It has been reported that the dendrites of aPN1 innervate the AMMC region while the aPN1 axons project to the WED region (*Vaughan et al., 2014*), indicating that the song signal is transmitted from AMMC to WED. To further investigate the directionality of information flow in the aPN1-vPN1-pC1 pathway, we have used DenMark and Syt::GFP for post- and pre-synaptic labeling of vPN1 and pC1 neurons, respectively. The dendrites of vPN1 target the WED region, and vPN1 axonal termini ramify extensively in the LPC region (*Figure 8A*). Both dendrites and axons of pC1 are observed within the LPC region (*Figure 8B*), and vPN1 axons overlap with pC1 dendrites when registered in a standard brain (*Figure 8C*). These results suggest that vPN1 neurons may convey song information from the WED to the LPC region.

To assess if the hypothesized pathway is indeed functional, we activated aPN1 or vPN1 with CsChrimson while recording GCaMP6-indicated calcium responses in vPN1 or pC1, respectively. CsChrimson was expressed in aPN1 by *R59C10-LexA::p65* and GCaMP6m was expressed in vPN1 by *R72E10-GAL4*. Pre-synaptic activation of aPN1 elicited calcium responses in vPN1 (*Figure 8D*),

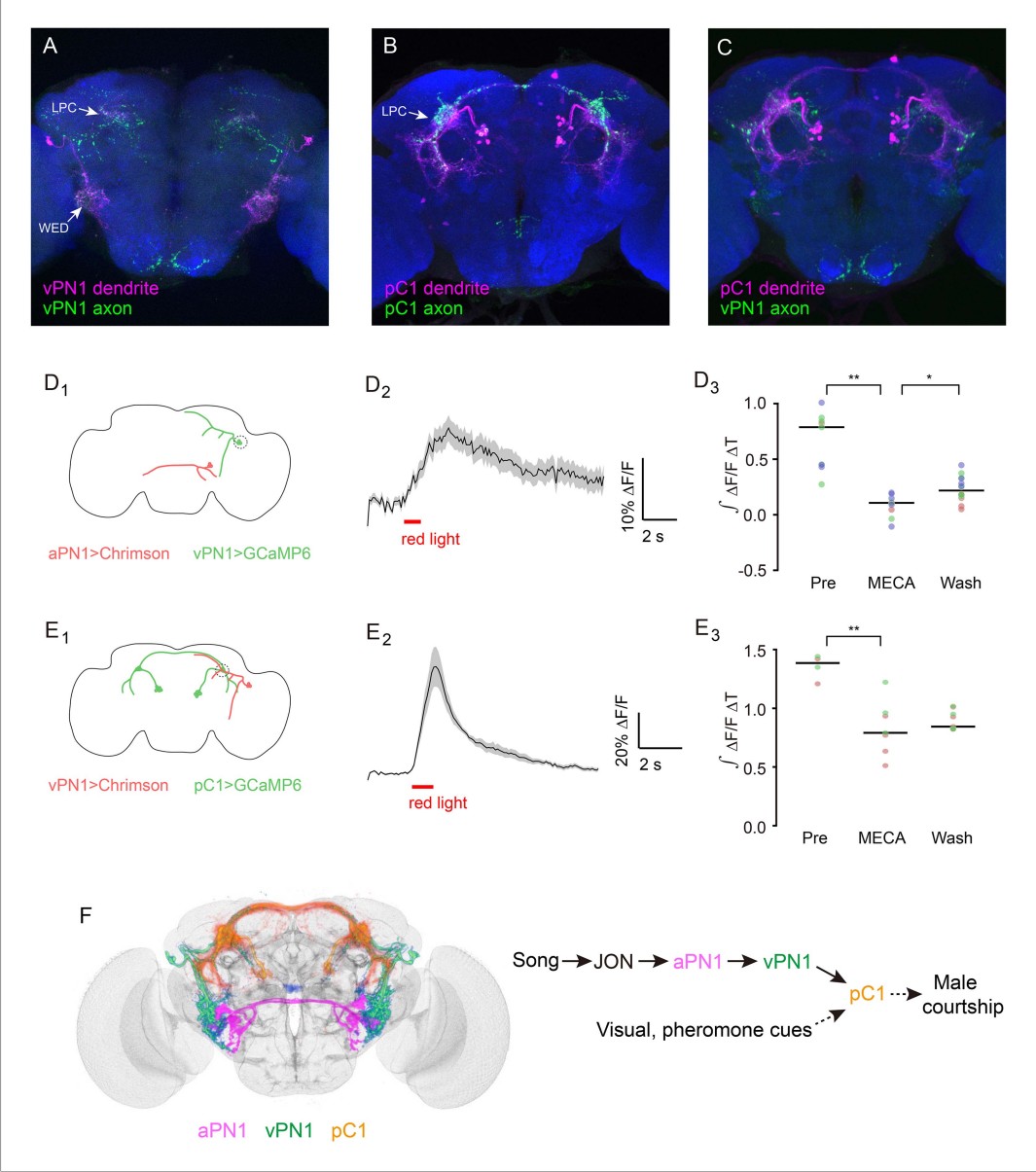

**Figure 8**. Directionality and functional connectivity of the auditory pathway. (**A–C**) Labeling of dendrites and axons by expression of the dendritic marker DenMark (magenta) and axonal marker Syt::GFP (green) in (**A**) vPN1 neurons (labeled with split GAL4 driver *R72E10-GAL4AD ∩ VT009665-GAL4DBD*) and (**B**) pC1 neurons (labeled with split GAL4 driver *R71G01-GAL4AD ∩ R15A01-GAL4DBD*). (**C**) Co-registration of vPN1 axons (green) and pC1 dendrites (magenta) onto a standard brain. (**D**) Activation of aPN1 neurons induced calcium responses in vPN1 neurons. (**D₁**) CsChrimson was expressed in aPN1 neurons with *59C10-LexA*, while GCaMP6 was expressed in vPN1 neurons with *R72E10-GAL4*. Dashed circle indicates the location of the recording site. (**D₂**) aPN1 CsChrimson-mediated activation induced calcium responses in vPN1 neurons. Black line indicates mean while gray envelope indicates SEM. n = 46 from 6 flies. (**D₃**) This effect is suppressed by the acetylcholine receptor antagonist mecamylamine (**p < 0.001, Student's *t*-test), and partially restored by washing out the antagonist (*p < 0.01, Student's *t*-test). Dots with the same color represent experiments performed on the same individual. Black lines indicate mean values. n = 9–14 (depending on the drug condition) from 3 flies. (**E**) Activation of vPN1 neurons induced pC1 calcium responses. (**E₁**) CsChrimson was expressed in vPN1 neurons with a vPN1 split-GAL4 driver while GCaMP6 was expressed in pC1 neurons with *dsx^LexA*. Dashed circle indicates the location of the recording site. (**E₂**) pC1 neurons respond to CsChrimson activation of vPN1 neurons. Black line indicates mean while gray envelope indicates SEM. n = 13 from 3 flies. (**E₃**) Mecamylamine causes a mild reduction in the pC1 responses (**p < 0.001, Student's *t*-test). Black lines indicate mean values. n = 4–9 from 2 flies. (**F**) Left panel shows co-registration of aPN1 (magenta), vPN1 (green), and pC1 (yellow) neurons onto a standard brain. The anatomical overlap between them suggests

*Figure 8. continued on next page*

*Figure 8. Continued*

a potentially interconnected circuit mediating courtship song detection in the male brain. We propose that courtship song is relayed through aPN1 and vPN1 neurons to pC1 neurons, and that the pC1 neurons integrate song signals with other sensory cues to initiate courtship.

The following figure supplement is available for figure 8:

**Figure supplement 1**. Characterization of synaptic connections in the aPN1-vPN1-pC1 pathway with GFP reconstitution across synaptic partners (GRASP) method.

suggesting a functional connection between aPN1 and vPN1. Moreover, to allow simultaneous vPN1 activation/pC1 recording, we generated a $dsx^{LexA}$ knock-in for driving GCaMP6m expression in $dsx+$ neurons while using $R72E10$-GAL4AD ∩ $VT9665$-GAL4DBD for CsChrimson-mediated activation of vPN1 neurons. As anticipated, activation of vPN1 elicited robust calcium responses in pC1 neurites (*Figure 8E*), supporting the idea that vPN1 neurons form functional synapses with pC1 neurons. In both cases, these effects were reduced by the administration of the acetylcholine receptor antagonist mecamylamine (*Figure 8D,E*). These data further support the existence of an aPN1-vPN1-pC1 functional pathway that mediates song detection in males (*Figure 8F*).

To further investigate the connectivity in the aPN1-vPN1-pC1 pathway, we first performed the GFP reconstitution across synaptic partners (GRASP) analysis to visualize the connections between aPN1, vPN1, and pC1 neurons. In this approach, two different populations of neurons were driven by either GAL4 or LexA (*Figure 8—figure supplement 1A–C*) to separately express each half of the GFP molecule (spGFP11 and spGFP1-10). GFP will be reconstituted in the region where these two groups of cells come into close proximity and form synapses (*Feinberg et al., 2008*; *Gordon and Scott, 2009*). By expressing spGFP11 in aPN1 neurons and spGFP1-10 in vPN1 neurons, we observed significant GRASP signal in the WED region (*Figure 8—figure supplement 1D*), suggesting the connections between aPN1 and vPN1. However, no GRASP signal was detected when we drove spGFP1-10 expression in vPN1 neurons with *72E10-GAL4* and spGFP11 expression in pC1 neurons with *71G01-LexA* (*Figure 8—figure supplement 1E*). One possibility to explain this is that *71G01-LexA* only label a subset of pC1 neurons, and given that pC1 neurons are likely to be heterogeneous (*Zhou et al., 2014*), the *71G01-LexA*-labeled pC1 neurons may not include the pC1 auditory neurons that synapse with vPN1 neurons. Alternatively, it might be that the neurites of vPN1 and pC1 in the LPC region is too diffuse to allow reliable detection of the GRASP signal. The mono-synaptic nature of this circuit is still awaiting a validation from electron microscopy or eletrophysiology methods.

## Discussion

Courtship behavior of *Drosophila* males provides a fundamental model for understanding how species-specific courtship signals may be processed and integrated to drive stereotyped motor outputs. Using anatomical, behavioral, and physiological approaches, here we outline a male-specific pathway for courtship hearing, which processes and transforms song stimuli to activate central $fru^M+$ or $dsx+$ neurons that support multimodal integration and drive courtship behavior.

### $fru^M$ neurons establish a labeled line that detects courtship song in males

*fru* and *dsx* are two key transcription factors with restricted expression patterns that specify the potential for sexual behaviors in *Drosophila* (*Manoli et al., 2013*; *Pavlou and Goodwin, 2013*; *Yamamoto and Koganezawa, 2013*). $fru^M$ expression in primary auditory, tactile, gustatory, and visual neurons as well as the central brain

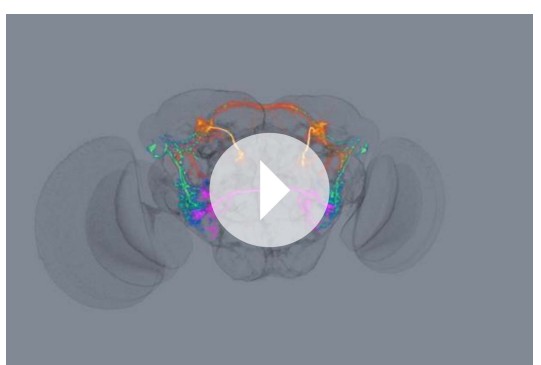

**Video 3.** 3D segmentation and co-registration of aPN1, vPN1, and pC1 neurons.

may suggest that there are multiple $fru^M$-labeled pathways conveying and integrating diverse sensory signals related to courtship, and an appealing hypothesis is that $fru^M$ labels interconnected neurons in a circuit that is dedicated to courtship (*Manoli et al., 2005*; *Stockinger et al., 2005*). For example, the male-specific pheromone cVA is processed by a four-neuron pathway extending from sensory neurons through to the ventral nerve cord (*Ruta et al., 2010*). This circuit appears to function as an olfactory labeled line, in that neurons in this circuit are functionally connected and selectively responsive to cVA.

We focused our efforts here on elucidating the auditory pathway underlying courtship song perception. We have demonstrated that the aPN1-vPN1-pC1 pathway is a labeled line for courtship hearing, by fulfilling four criteria: (1) these neurons are functionally connected; (2) these neurons respond preferentially to courtship song; (3) these neurons are necessary for the behavioral response to courtship song in male flies, and (4) activation of this labeled line provides a fictive stimulus, observable by the chaining response elicited upon CsChrimson activation. Strikingly, this labeled line appears to be specified by the expression of $fru^M$ or $dsx$.

The auditory labeled line for courtship hearing begins with $fru^M$-expressing JONs and second-order auditory neurons aPN1/aLN(al) in the AMMC (*Manoli et al., 2005*; *Stockinger et al., 2005*; *Vaughan et al., 2014*). Silencing either $fru^M$ JONs or $fru^M$ aPN1 neurons reduced male song-induced responses (*Figure 4*) (*Vaughan et al., 2014*). In addition, we have also demonstrated the connectivity, response patterns, necessity, and sufficiency of $fru^M$ vPN1 and pC1 neurons in this pathway, thus, delineating a labeled line of $fru^M$ neurons leading directly from sensory neuron to multimodal integration.

The inclusion of vPN1 in this pathway is supported by three lines of evidence. First, vPN1 projections extensively overlap with projections extended by aPN1 neurons within WED, and we observe functional connectivity between aPN1 and vPN1. Second, silencing vPN1 reduced pulse song-induced chaining in male flies, while optogenetic activation of vPN1 neurons mimicked a song signal to induce male chaining. Third, GCaMP recordings reveal that vPN1 responds strongly to both pulse song and sine song. We therefore conclude that $fru^M$+ vPN1 neurons are the third-order neurons mediating courtship hearing.

vPN1 may provide its output via innervation of the LPC, a region receiving multimodal input that is likely to be a site for multi-sensory integration (*Yu et al., 2010*). This area is heavily innervated by $dsx$+ pC1 neurons, which include most of the male-specific $fru^M$+ P1 neurons (*Kohatsu et al., 2011*; *Zhou et al., 2014*). While the broader pC1 population is important for both male courtship and female receptivity, the P1 neurons play a critical role in the initiation of male courtship and respond to both male and female pheromones (*Kohatsu et al., 2011*; *von Philipsborn et al., 2011*; *Pan et al., 2012*; *Zhou et al., 2014*). These neurons appear to be the downstream targets of vPN1, based on three lines of evidence. First, the arborizations of pC1 neurons match very closely with the projection of vPN1 neurons in the LPC, and optogenetic activation of vPN1 generates robust activity in pC1. Second, pC1 neurons show calcium responses to pulse song stimuli, with IPI tuning that matches that of the behavioral response. Third, silencing pC1 neurons in male flies almost completely abolishes song-induced chaining, while activation induces robust chaining in the absence of song. We therefore conclude that vPN1 may carry song stimuli to activate pC1, where these stimuli are integrated with other sensory modalities such as pheromonal olfactory and gustatory cues to modulate the courtship level in males (*Figure 8F*).

Taken together, the neural circuit we identified suggests that song information flows via a labeled line of $fru^M$ neurons from the antenna to AMMC, to WED, and then to LPC, providing a functional explanation of how pulse song induces male courtship behavior.

## IPI tuning of auditory pathway

IPI is a key parameter of courtship song that exhibits great variation across *Drosophila* species (*Cowling and Burnet, 1981*). *D. melanogaster* not only produces song with a specific IPI (*Shorey, 1962*; *Bennet-Clark and Ewing, 1967*; *von Philipsborn et al., 2011*), but also behaviorally recognizes song with that conspecific IPI in both males and females (*Bennet-Clark, 1969*; *Yoon et al., 2013*). We have also shown that song-induced male-chaining behavior is most responsive to a 35-ms IPI, although longer IPIs (35–65 ms) are still able to induce robust chaining behavior.

While *Drosophila* has behavioral preferences toward the conspecific IPI, it has not been clear how IPIs are represented in the nervous system or how the fly discriminates specific IPIs. Our results suggest there is a significant change in pulse song representation across the ascending aPN1-vPN1-pC1 pathway. For aPN1, the GCaMP $\Delta$F/F responses in female flies reflect an integration of pulse rate at IPIs longer than 25 ms (*Vaughan et al., 2014*). In contrast, vPN1 responses observed here are low-passed

and preferentially tuned to longer IPIs. Interestingly, the vPN1 response saturates above ~35-ms IPI, consistent with the saturating response observed when comparing dendritic and axonal GCaMP signals in aPN1 (*Vaughan et al., 2014*). Notably, however, neither the aPN1 nor vPN1 response corresponds well with the behavioral sensitivity to IPI observed in male or female flies.

In contrast, the IPI sensitivity of pC1 reflects a band-pass response to IPI that closely matches the behavioral sensitivity of the chaining response. Indeed, the correlation between pC1 response and chaining behavior is significantly higher than the correlation observed for vPN1. Thus, while the mechanistic details remain unclear, the IPI sensitivity appropriate for species-appropriate responses is likely to be generated through a multi-stage transformation of song stimuli.

## Sexual dimorphism of auditory circuits

Sexual dimorphism at multiple levels in the *Drosophila* brain may give rise to sex-specific differences in sensory processing and multimodal integration. The central integrators of courtship-related sensory cues in male and female flies, the pC1 neurons, are themselves sexually dimorphic in both cell number and morphology (*Kimura et al., 2008*; *Kohatsu et al., 2011*; *Zhou et al., 2014*). pC1 neurons arborize within the triangular lateral junction of the LPC in both sexes, where integration of multiple sensory modalities may occur, but they also show male-specific innervation of the LPC arch and male-specific contralateral projections (*Kimura et al., 2008*; *Zhou et al., 2014*).

For courtship hearing, pC1 neurons are stimulated by pulse song in both sexes, but are also stimulated by sine song in females (*Zhou et al., 2014*). This result is consistent with the behavioral observation that both males and females are responsive to pulse song, while females are also responsive to sine song (*Bennet-Clark, 1969*; *Schilcher, 1976a*, *1976b*; *Eberl et al., 1997*; *Shirangi et al., 2013*). However, the pC1 auditory response cannot be easily explained by the dimorphism of vPN1, which responds to pulse song in males but is absent in females. Moreover, the absence of vPN1 in females begs the question of how pC1 receives song information in females. One explanation comes from the observation that vPN1 is a subset of the *fru+* aSP-k clone (*Cachero et al., 2010*). aSP-k shows arborization in VLP and the LPC ring in both male and females, as well as male-specific innervation of the LPC arch that corresponds with vPN1 morphology. These neurons, including non-*fru+* neurons in the same lineage, may compose a parallel pathway for female hearing.

More generally, we observe a gradient of sexual dimorphism across the ascending pathway for both olfaction and audition. In both cases, we note only limited sexual dimorphism in second-order neurons (DA1 and aPN1, respectively), but dramatic changes in third-order neurons (aSP-f/aSP-g and vPN1) and integrative neurons (pC1), which show significant dimorphisms in cell number and morphology (*Datta et al., 2008*; *Ruta et al., 2010*; *Kohatsu et al., 2011*; *Kohl et al., 2013*; *Zhou et al., 2014*). This gradient may reflect a general rule for the flexible assembly of sexually dimorphic circuits on an evolutionary timescale.

Our anatomical, behavioral, and physiological analyses here have outlined the architecture of a system supporting species-specific courtship hearing, built upon genetically labeled lines expressing *fru*$^M$ or *dsx* within the fly. Although it is clear that courtship song representations are systematically transformed along the aPN1-vPN1-pC1 pathway, we await a circuit and synapse-level explanation for how this occurs, as well as an understanding of how pC1 activation gives rise to distinct and appropriate behavioral outputs in each sex.

# Materials and methods

## Fly stocks

The *fru*$^{LexA}$ and *dsx*$^{GAL4(\Delta2)}$ lines were previously described (*Mellert et al., 2010*; *Pan et al., 2011*). CRM-GAL4s, R72E10-GAL4AD, R71G01-GAL4AD, R15A01-GAL4DBD, LexAop2-FLP (pJFRC79 in *attP40*), LexAop2-FLP (pJFRC79 in *attP2*), UAS>stop>myr::GFP (pJFRC41 in *attP40*), UAS>stop>myr::GFP (pJFRC41 in *su(hw)attP1*), and 20XUAS>dsFRT>CsChrimson-mVenus (in VK00005) were gifts from Gerald Rubin (Janelia Research Campus) (*Pfeiffer et al., 2008*; *Jenett et al., 2012*). 22B11-GAL4 is previously reported (*Vaughan et al., 2014*). R71G01-LexA::p65 (inserted in *attP40*) was previously described (*Pan et al., 2012*). UAS-GCaMP6m was a gift of Douglas Kim (Janelia Research Campus). *fru*$^M$, VT9665-GAL4DBD, UAS>stop>TNT, and UAS>stop>TNT$^{in}$ were gifts from Barry Dickson (Janelia Research Campus). LexAop-spGFP11 and UAS-spGFP1-10 were gifts from Kristin Scott (UC Berkeley). UAS-DenMark, UAS-syt::GFP was previously described (*Nicolai et al., 2010*).

## Immunohistochemistry

5- to 7-day-old adult fly CNSs were dissected in Schneider's insect medium (Sigma, MO, United States, S0146) and immediately fixed in 2% paraformaldehyde (PFA) in Schneider's insect medium for 55-min at room temperature (RT). After washing three times with PBS (Phosphate-buffered saline) containing 0.3% Triton X-100 (PBT), the samples were incubated in PBT with 5% normal goat serum (Vector Laboratories, CA, United States) for 1 hr at RT. Samples were then incubated at 4°C for 24 hr in primary antibody, washed three times in 0.3% PBT at RT, and then incubated at 4°C for 24 hr in secondary antibody. After washing three times with 0.3% PBT, samples were fixed in 4% paraformaldehyde (PFA) for 3 hr at RT, washed again five times with 0.3% PBT, and mounted onto a poly-lysine-coated coverslip. The coverslip was then immersed into 30%, 50%, 75%, 95%, 100% ethanol for dehydration at 5-min intervals. Next, the coverslip was washed with Xylenes (Fisher Scientific, NJ, United States) for three times in the hood and mounted in DPX solution (Electron Microscopy Sciences, PA, United States) before imaging. Primary antibodies were mouse anti-Bruchpilot nc82 (Developmental Studies Hybridoma Bank, IA, United States) used at a 1:50 dilution, Fru$^M$ antibody used at 1:100 dilution, and rabbit anti-GFP used at a 1:1000 dilution (Invitrogen). Secondary antibodies were Alexa Fluor 546 goat anti-mouse IgG and Alexa Fluor 488 goat anti-rabbit IgG (Invitrogen, OR, United States) used at a 1:500 dilution.

## Brain registration

A standard brain was generated using CMTK software to average six male and female brains stained with nc82 (anti-Bruchpilot) that were of good quality (*Rohlfing and Maurer, 2003*; *Rohlfing, 2012*). Confocal stacks were registered into the standard brain by linear registration and non-rigid warping based on the nc82 channel (*Jefferis et al., 2007*; *Ostrovsky et al., 2013*).

## Song-induced chaining behavior

Flies were reared at 22°C and 50% humidity under a 12 hr:12-hrlight/dark cycle. Male flies were collected immediately after eclosion and aged in groups of 8 for 2 days. They were then anesthetized to remove their wings and allowed to recover for 5–7 days before assaying the behavior. Behavioral experiments were performed at 22°C. Briefly, six males were introduced into the chaining chamber by gentle aspiration without anesthesia, and then videotaped by a Stingray camera (F080B, Allied Vision Technologies, PA, United States) for 6-min. After 1-min in the chamber, continuous pulse song trains were played back with incremental intensities at 30-s intervals. Two pieces of nylon mesh were inserted at both ends of the chaining chamber to allow the delivery of acoustic stimuli from an external speaker (HiVi D10G Woofer, Parts Express, Springboro, OH) located 9-cm from the center of the chamber.

Synthetic pulse song was generated by MATLAB as a train of Gaussian-modulated sinusoidal pulses with a 220 Hz fundamental carrier frequency and a defined IPI. A data acquisition device (USB-6229 BNC, National Instruments) controlled by a custom-written MATLAB program (*Source code 1*) was used to relay the song signals to an A500 linear amplifier (Willich, Germany) that drives the speaker, triggers the videotaping, and maintains synchrony between the two.

To calculate the chaining index, the number of flies engaged in chaining was counted every 3 s and summed up for each 30-s block. The maximum chaining index for one block is 60 if all six flies are chaining throughout the 30-s block. Furthermore, a detailed analysis of chaining behavior for each group of flies was presented in a heat map to reflect the number of chaining flies every 3 s throughout the recording time.

## Calcium imaging

*R72E10-GAL4/UAS-GCaMP6m* and *dsx$^{GAL4}$/UAS-GCaMP6m* males were aged in groups of 8–12 for 5–7 days before calcium imaging. Flies were gently anesthetized on ice and then inserted into a rectangular hole (~1 mm × 2 mm) and stabilized with low-melting wax as described previously (*Zhou et al., 2014*). The dorsal head capsule was facing up to the objective and bathed in insect saline (103 mM NaCl, 5 mM HEPES, 8 mM trehalose, 10 mM glucose, 26 mM NaHCO$_3$, 1 mM NaH$_2$PO$_4$, 2 mM CaCl$_2$, 1.5 mM MgCl$_2$, pH = 7.3), while the antenna protrudes downward on the other side of the plate to receive acoustic stimulation. Because *R72E10-GAL4* labels off-target neurons with projections intermingled with the neurites of vPN1 neurons, it is technically difficult to

record calcium responses from the vPN1 neurites. We therefore performed calcium imaging from the somas of vPN1 neurons, which are easily identified nearby the lateral horn. The cuticle at the dorsal region of the head was gently removed with sharp forceps, and calcium signals from the somas of vPN1 neurons were recorded. For imaging the neurites of male pC1 neurons, the procedure was the same as previously described (*Zhou et al., 2014*). At the end of experiments, most flies appeared healthy as they were still exhibiting voluntary abdomen contractions.

Calcium-imaging experiments were done with a 488-nm laser on a Zeiss LSM 710 confocal microscope. Images with 128 × 128 pixels resolution at a frame rate of 13 Hz were acquired with a water immersion objective lens (40×/1.0 DIC VIS-IR, Zeiss). A data acquisition device (USB-6229 BNC, National Instruments) was used to control a trigger interface box (1437-440, Zeiss) and an amplifier (Marantz SR5003) to synchronize song stimulus onset and image acquisition. An external speaker located ~20-cm from fly antenna was used to present the song stimuli.

Synthetic song stimuli including pulse song (40 pulses) with a defined IPI, 1.4-s sine song (140 Hz) and 1.4-s white noise were generated with MATLAB software. The sound intensity was measured with a sound level meter (NO.33–2055, Radioshack). We also measured the synthetic song with a particle-velocity microphone (Microflown Technologies, Arnhem, NL) to make sure that the song intensity (80 dB) used in calcium imaging is within the range of natural *Drosophila* courtship song (*Figure 6—figure supplement 3*). To reduce the habituation effects, song stimuli were shuffled randomly and presented at 1-min intervals.

## CsChrimson activation

52 high-power red LEDs (655 nm, LXM3-PD01, Luxeon Rebel) and 96 blue LEDs (468 nm, VAOL-S12SB4, VCC Optoelectronics, CA, United States) were mounted onto a heat sink and suspended above the chaining chambers to provide a source of illumination. Blue LEDs provide constant background illumination to allow flies to see each other, as male chaining behavior depends on the ability to visually tracking other flies. Because CsChrimson is also sensitive to blue light, we kept the blue light at a very low level (0.001 mW/mm$^2$) to make sure that no chaining behavior was triggered with only blue light illumination. The intensity and frequency of high-power red LEDs were controlled with a Teensy 2.0 microcontroller using MATLAB software. Light intensity was measured by placing an optical power meter (PM100D, Thorlabs, NJ, United States) nearby the location of chaining chambers. Fly behavior was recorded by a Stingray camera equipped with a Tokina infrared filter under 880-nm LED illumination (SL1236, Advanced Illumination, VT, United States) to avoid the interference from red and blue LEDs.

For all the CsChrimson experiments, crosses were set up on standard fly food with 0.2 mM all-trans-retinal. Male flies were collected immediately after eclosion and reared in groups of 8–12 on 0.5 mM retinal food for 5–7 days before behavioral test. A group of eight males with the same genotype were gently aspirated into a culture dish (430165, 35 mm × 10 mm, Corning, NY, United States) containing fly food at the bottom. Videotaping and red light stimulation were triggered simultaneously with a MATLAB interface for a total duration of 5 min. Videos were scored using LifesongX software. The percentage of time when at least three flies are in a courtship chain was calculated to quantify the chaining behavior induced by CsChrimson activation.

## Generation of *dsx*$^{LexA::p65}$

To genetically label cells that express *dsx*, we used homologous recombination (*Gong and Golic, 2003*) to replace the entire coding sequence of dsx exon 2 (starting after the ATG through the splice donor) with the coding sequence for *LexA::p65* (*Pfeiffer et al., 2010*) followed by a stop codon and a transcription stop cassette containing the SV40 poly-A sequence in tandem with the *D. melanogaster* α-tubulin 84B 3′ UTR (*Stockinger et al., 2005*). For homologous recombination, we generated donor transgenes in which these exogenous sequences were flanked by two homology arms corresponding to the *dsx* locus: a 2.8-kb 5′ homology arm as per (*Robinett et al., 2010*) and a 2.7-kb 3′ homology arm extending between genomic sequences GCAATATTGGCACTCAGCTATTATC and CACGTTCGA TATTGAGTTGGGTGAA in the *dsx* second intron. The 3′ arm was PCR-amplified from genomic DNA prepared with the DNeasy Tissue Kit (Qiagen). All DNA fragments were generated by PCR using AccuPrime Supermix (Invitrogen) and sequence-verified. Using restriction endonuclease sites added to the 5′ ends of the PCR primers, these fragments were cloned in the linear order of *dsx* 5′

arm-*LexA::p65-SV40 poly-A/α-tubulin 84B 3′ UTR-dsx 3′* arm in pBluescript-SK (Invitrogen) and then transferred as a unit into *pP{WhiteOut2}* (gift of Jeff Sekelsky) to make *pP{WO2-dsx-LexA::p65-stop-2}*.

*pP{WO2-dsx-LexA::p65-stop-2}* transgenics were made by P element-mediated germ line transformation using standard methods (Rainbow Transgenic Flies, Inc.), and nine independent, homozygous viable, non-third chromosome transformant lines were isolated to serve as donors of the *dsx-LexA::p65-stop-2* DNA fragment for homologous recombination (*Gong and Golic, 2003*). Donors were crossed to a line with heat shock-inducible FLP recombinase and I-SceI endonuclease transgenes (*Gong and Golic, 2003*), and larvae were heat shocked for 1 hr at 37°C on days 3 and 4 of development. ~7000 female F1 progeny containing the two transgenes and the donor were crossed to *lexAop-rCD2::GFP* (*Lai and Lee, 2006*) males and the F2 progeny screened for candidates with a GFP expression pattern matching expression of *dsx^GAL4(1)* (*Robinett et al., 2010*). 40 independent candidate flies were isolated, but only six were fertile. Of these candidate lines, four produced intersexual progeny when homozygosed or when heterozygous with *Df(3R)dsx^{M+R15}* (*Baker et al., 1991*). These four lines were tested by PCR for proper targeting of *LexA::p65-stop* into the endogenous *dsx* locus by using the *dsx* 5′ genomic and *LexA::p65* primers, GTGTGTGAGGCTGCC TATGTACTAG and GACACGATTTCAATGACACCCTTGC, respectively, and the *dsx* 3′ genomic and *α-tubulin 84B 3′ UTR* primers, GAAAGTCGCAGTTTCCTACTGATAC and CCGTCAAGCATGC GATTGTACATAC, respectively. For each candidate, the predicted 5′ and 3′ PCR products were generated, confirming proper targeting.

## Functional connectivity

Male flies 5 to 7 days old were dissected in saline containing: 103 mM NaCl, 3 mM KCl, 5 mM TES, 8 mM trehalose dihydrate, 10 mM glucose, 26 mM NaHCO$_3$, 1 mM NaH$_2$PO$_4$, 2 mM CaCl$_2$, 4 mM MgCl$_2$, bubbled with carbogen (5% CO$_2$/95% O$_2$). The brain and ventral nerve cord were taken out of the fly and laid on a poly-lysine-coated coverslip. The dissection was realized using the minimum level of illumination possible to avoid spurious activation of CsChrimson. Brains were then continuously perfused in the same saline at 60 ml/hr throughout the experiment. Imaging was done on a two-photon scanning microscope (PrairieTechnologies, Bruker). Excitation wavelength was 920 nm. CsChrimson was excited with 50-Hz trains of 2-ms 590-nm light pulses via a LED shining through the objective. Instantaneous powers measured out of the objective ranged between 50 μW/mm$^2$ and 700 μW/mm$^2$. Experiments usually started with a 50 pulses train at 50 μW/mm$^2$. If no response could be seen, the power was raised progressively until a response could be seen or the maximum power was reached. For pharmacology experiments, mecamylamine (50 μM) was administered through the perfusion line for times ranging from 3 to 5 min, followed by a wash period where the perfusion was drug-free again.

For aPN1 activation/vPN1 recording, *R59C10-LexA::p65/LexAop2-CsChrimson; R72E10-GAL4/ UAS-GCaMP6m* flies were used and the cell soma of vPN1 neurons was recorded, as it was the only place where we could confidently identify the vPN1 neurons. For vPN1 activation/pC1 recording, *R72E10-GAL4AD/UAS-CsChrimson; VT9665-GAL4DBD, dsx^{LexA}/LexAop2-GCaMP6m* flies were used and the LPC projection of pC1 neurons was imaged.

In the cases where cell somas were imaged (vPN1 imaging), fluorescence videos were segmented based on average intensity using k-means clustering, following which individual cell bodies were labeled as different region of interests.

In the cases where LPC projection was imaged, fluorescence videos were segmented by k-means (with 3 clusters) on the individual pixel traces of every run.

In both cases, ROIs were then cleaned up by an erosion/dilation step.

ΔF/F were then calculated for each run with F being the average signal before the stimulation. For further analysis, only the most responding ROI was kept.

## Data analysis

Statistical analysis was performed with MATLAB software. For the song-induced chaining assay, Wilcoxon rank-sum test was used to detect significant differences between TNT and control groups at 80 dB song playback.

For GCaMP data, ROI selection and peak ΔF/F calculation were done using custom-written code in MATLAB (*Source code 2*). Calcium traces were smoothed with a cubic Savitzky-Golay filter. Peak ΔF/F was calculated from $\Delta F/F = (F_t - F_b)/F_b$, where $F_t$ is the maximum value in the 5-s window following stimulus onset and $F_b$ is the averaged value in the 1-s window before the stimulus onset.

Wilcoxon signed-rank test was used to compare vPN1/pC1 responses at 35-ms IPI to those at shorter or longer IPIs (*Figures 5H, 6K*). To normalize responses, each fly's response across IPIs was divided by its maximum response; these responses were subsequently averaged to generate the values presented in *Figure 6L–N*. To assess the correlation between calcium responses and chaining behavior with respect to IPI tuning, we calculated the Pearson correlation coefficient, $r$, between GCaMP and behavioral responses normalized by the maximum response value. The significance of this correlation was established by calculating a bootstrap distribution of correlations between samples with randomly permuted IPIs. The vPN1/behavior correlation was compared to the pC1/behavior correlation via Meng's z test (*Meng et al., 1992*). The significance of the tuning curve shown in *Figure 6—figure supplement 2A* was calculated using a one-sided Wilcoxon rank-sum test.

## Acknowledgements

We thank Gerald Rubin, Douglas Kim, and Barry Dickson for providing fly lines; Troy Shirangi, Dave Mellert, Kristine Branson, Jue Xie, Yingxue Wang, and members of the Baker lab for helpful discussions; Peter Polidoro for constructing the courtship chamber; Tanya Tabachnik, Steven Sawtelle, Jinyang Liu, Gus Lott, and Dominik Hoffmann for engineering support; Ming Wu and Yoshi Aso for helps with LED setup; the Janelia Research Campus Fly Core for technical assistance; and Alison Howard for administrative support. This work was supported by the Howard Hughes Medical Institute.

## Additional information

### Funding

| Funder | Author |
| --- | --- |
| Howard Hughes Medical Institute (HHMI) | Bruce S Baker |

The funder had no role in study design, data collection and interpretation, or the decision to submit the work for publication.

### Author contributions

CZ, Conception and design, Acquisition of data, Analysis and interpretation of data, Drafting or revising the article; RF, Acquisition of data, Analysis and interpretation of data; AGV, Analysis and interpretation of data, Drafting or revising the article; CCR, Drafting or revising the article, Contributed unpublished essential data or reagents; VJ, Conception and design, Analysis and interpretation of data; BSB, Conception and design, Analysis and interpretation of data, Drafting or revising the article

## Additional files

### Supplementary files

• Source code 1. A Matlab GUI for synchronized video-taping and song stimulation in *Figure 3*.

• Source code 2. A Matlab GUI for processing and analyzing the calcium-imaging data in *Figure 5*.

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
