## [Decision Letter]

Thank you for submitting your work entitled “Central Neural Circuitry Mediating Courtship Song Perception in Male *Drosophila*” for peer review at *eLife*. Your submission has been favorably evaluated by K VijayRaghavan (Senior Editor) and three reviewers, one of whom is a member of our Board of Reviewing Editors. The following individuals responsible for the peer review of your submission have agreed to reveal their identity: Mani Ramaswami (Reviewing Editor) and Dan Eberl and Ralph Greenspan (peer reviewers).

The reviewers have discussed the reviews with one another and the Reviewing Editor has drafted this decision to help you prepare a revised submission.

This manuscript describes a substantial effort to understand the neural pathway underlying the male version of the sexually dimorphic response to courtship song. Zhou et al. identify third and fourth order neurons in the fly brain that respond to courtship song. They beautifully demonstrate the connectivity of the neurons, their responsiveness to sound, and their involvement in male courtship behavior. Strikingly these synaptically connected neurons are both necessary and sufficient to mediate the male response to courtship song. The manuscript is well written, the experiments well executed and the conclusions are appropriate and make an exciting advance to the field. This work parallels another recent publication by the same group looking at the female equivalent pathway.

Points to be addressed in a revised manuscript:

1) The conclusion that the neurons are synaptically connected would be strengthened by GRASP experiments. Why are these not attempted or reported? If it is a formal possibility that the connections are indirect, then this should be stated.

2) The conclusion that vPN1 show low-pass filtering and pC1 neurons show band-pass filtering appears to be based on rather slim data. It could well be true but, particularly given the very weak response of pC1 neurons, is it possible that differences between these two neuronal types could be minor and that there could well be no real transformation of song representation from one to the other? This conclusion should be appropriately strengthened or qualified.

3) The authors mention that song induced male chaining is most responsive to ∼35ms IPI and that aPN1 integrates the pulse rate and vPN1 is a low pass filter. I couldn't seem to find any data or discussion of aPN1 or other projection neurons being able to integrate pulse rates faster than 35ms IPI.

4) Why does aPN1 activation not induce chaining behavior, if activation of the downstream neurons do? Is this a technical problem: i.e. not sufficient activation by the Chrimson system? Or is there a more interesting possibility? This should be acknowledged or discussed.

5) What if it's not vPN1 acting as a low pass filter, but 35ms being the natural limit of the fly auditory system and in turn aPN1 integration? Are there other circuits tuned for higher frequencies? If so, it's worth mentioning. This could potentially argue as another reason for how pC1 in females perceives song information despite lacking the so-called low pass vPN1 filter.

6) In the second paragraph of the subsection “Song-induced chaining behavior is tuned to conspecific IPIs”, this tuning would be superimposed over the receiver tuning in the JO itself in different species ([46], Curr. Biol.).

7) In the first sentence of the fourth paragraph of the Introduction, the word “perceive” here is incorrect. “Receive” is more appropriate here. In other places in the manuscript the authors correctly use the word “perceive” in the context of decoding the sensory signals.

8) In the fourth paragraph of the Introduction, upon mention of the projections to AMMC, the authors should mention that the auditory JO neurons also have a direct input to the Giant Fiber. Relevant citations are Lehnert et al., which they already cite elsewhere in a different context, and [43], J Neuroscience.

---

## [Author Response]

1) The conclusion that the neurons are synaptically connected would be strengthened by GRASP experiments. Why are these not attempted or reported? If it is a formal possibility that the connections are indirect, then this should be stated.

We thank the reviewers for this suggestion. We have now conducted GRASP experiments (Figure 8—figure supplement 1) and found that reconstituted GFP signals are present between aPN1 and vPN1 neurons but barely detectable between vPN1 and pC1 neurons. One possibility could be that the projections of vPN1 and pC1 neurons in the LPC region are relatively diffuse, posing a great challenge for GRASP detection. We should also note that the *71G01-LexA* driver used for GRASP only labels a small subset of pC1 neurons, while the intersection of *71G01-LexA* and *dsx*^*GAL4*^ actually labels a broader group of pC1 neurons. The higher number of pC1 neurons labeled by the intersectional approach is likely due to flip-out of the stop cassette during early developmental time-points that might transiently express*71G01-LexA*. Thus, *71G01-LexA* alone may not label the subset of auditory pC1 neurons that synapse with vPN1 neurons. Nonetheless, we acknowledge that our data do not prove that aPN1-vPN1-pC1 neurons are mono-synaptically connected, and we have carefully revised the text to reflect this.

*2) The conclusion that vPN1 show low-pass filtering and pC1 neurons show band-pass filtering appears to be based on rather slim data. It could well be true but, particularly given the very weak response of pC1 neurons, is it possible that differences between these two neuronal types could be minor and that there could well be no real transformation of song representation from one to the other? This conclusion should be appropriately strengthened or qualified*.

Thank you for raising these concerns. We now make textual changes to address these concerns.

First, we've toned down the overall statement regarding IPI transformation between vPN1 and pC1.

Second, we've changed the heading of the Results sub-section to “Potential transformation of song responses between vPN1 and pC1” to allow the possibility that the GCaMP response may suffer from some artifact.

Third, we've added a comment on the caveats of calcium imaging to the Results sub-section: “As caveats, however, we note that the slow decay kinetics of GCaMP6m (7) may account for some of the observed low-pass properties observed in our vPN1 or pC1 recordings. […] Indeed, although our results suggest that song representations may be serially transformed in the ascending auditory pathway, a circuit-level understanding of how local and projection neurons in this pathway shape this response remains to be discovered.”

*3) The authors mention that song induced male chaining is most responsive to ∼35ms IPI and that aPN1 integrates the pulse rate and vPN1 is a low pass filter. I couldn't seem to find any data or discussion of aPN1 or other projection neurons being able to integrate pulse rates faster than 35ms IPI*.

The aPN1 response to IPIs was reported previously (61) where Figure 7 shows that aPN1 response is proportional to pulse rate at IPIs longer than 25 ms. We apologize for this confusion and now cite this paper more clearly when we refer to the aPN1 results. In addition, we now include a template of the aPN1 response in (Figure 6—figure supplement 2) to allow for direct comparison.

4) Why does aPN1 activation not induce chaining behavior, if activation of the downstream neurons do? Is this a technical problem: i.e. not sufficient activation by the Chrimson system? Or is there a more interesting possibility? This should be acknowledged or discussed.

We believe that the insufficiency of Chrimson-activated aP1 neurons to induce male chaining is biologically relevant and is not a technical failure of the Chrimson system. The possibility that Chrimson system is not sufficient to activate aPN1 is less likely because we used multiple aPN1 drivers and relatively high LED power for red light stimulation. Moreover, the inability of aPN1 activation to induce male chaining is in line with the previous report that continuous [dTrpA1] activation of aPN1 neurons failed to restore female receptivity to wingless males (61).

Our leading hypothesis is that activation of aPN1 does not induce chaining for the reason that aPN1 may not encode song quality as a rate code. Evidence for this includes the observation that the aPN1 GCaMP response is maximal at an IPI (25 ms) that is not optimal to induce a behavioral response (see e.g. Figure 3). We think it is likely that the early temporal code in JON/aPN1 is transformed into a rate code within vPN1/PC1. While genetic activation is highly effective at evoking behavioral responses from presumably rate-coding neurons, it is insufficient to mimic the structure of temporal coding.

Moreover, we suspect that courtship song representation at the level of second-order neurons involves multiple neuronal cell types and solely activating aPN1 neurons is not sufficient to mimic a fictive courtship song stimulus. For example, aLN(al) neurons are GABAergic local inter-neurons that innervate AMMC and are behaviorally required for courtship hearing (61), however, it is still unknown how aLN(al) may communicate with aPN1 to process song signals. Alternatively, given the heterogeneous responses of AMMC-B1/aPN1 neurons to acoustic stimulus (Figure 2 in (30)), an uniform activation of aPN1 neurons may not be appropriate for behavioral induction.

Currently we favor the hypothesis that as song information flows from peripheral to central brain, the higher-order neurons may represent more salient and socially relevant features of courtship song than lower-order neurons, and this may account for the gradient of behavior phenotype we observed when we activate aPN1, vPN1 or pC1 respectively. We now discuss the behavioral differences upon Chrimson activation as follows:

“In particular we note that the lack of response for aPN1 is consistent with the observation that the stimulus that best activates aPN1 (25-ms IPI) does not induce a strong behavioral response from wild-type flies. […] This (still theoretical) architecture describes a possible transformation of song representations from a temporal code in aPN1 to a rate code in vPN1 – which is ultimately coupled with other sensory signals to encode overall behavioral arousal in PC1 (Figure 8 and Video 3).”

*5) What if it's not vPN1 acting as a low pass filter, but 35ms being the natural limit of the fly auditory system and in turn aPN1 integration? Are there other circuits tuned for higher frequencies? If so, it's worth mentioning. This could potentially argue as another reason for how pC1 in females perceives song information despite lacking the so-called low pass vPN1 filter*.

We thank the reviewers for raising this concern. Pulse songs have two key features: intra-pulse frequency (IPF) and inter-pulse interval (IPI). It was determined with laser doppler vibrometric measurements that the optimal frequency detection for the *D. melanogaster*'s antennal ear is ∼147 Hz, and in general the antennal receiver tuning is strongly correlated with the intra-pulse frequency of pulse songs in different *Drosophila* species rather than the inter-pulse interval (46). Moreover, previous study have revealed that the strongest responses in aPN1 arises at 25-ms IPI, well below the peak behavioral activation of 35-ms IPI (61).

While it is not possible to quantitatively compare the aPN1 responses in the previous study to vPN1 responses in this study (due to use of different GCaMP reagents and other experimental changes), it does seem clear that aPN1 responses and vPN1 responses are qualitatively different (Figure 6—figure supplement 2). However, it is certainly correct that the mechanism of low-pass filtering for vPN1 need not reside within vPN1, and that we should not attempt to claim this.

To make this more clear, we have altered the discussion of this result to say “This [vPN1] signal displays low-pass response properties for IPI with a shoulder around 35-ms IPI, and is different from the aPN1 response that responds as a function of pulse rate (61).”

*6) In the second paragraph of the subsection “Song-induced chaining behavior is tuned to conspecific IPIs”, this tuning would be superimposed over the receiver tuning in the JO itself in different species (*[46]*, Curr. Biol.)*.

As the authors mentioned before, receiver tuning is correlated with the IPF of conspecific pulse songs (46), while IPI tuning is more likely to be implemented by neural circuits in the brain. This possibility is also surmised by Riabinina et al., “if in the acoustic communication system of *Drosophila* it is the brain's prime task to detect conspecific pulse trains by analyzing the intervals between individual pulses, then it should be the ear's prime task to detect the individual pulses in the best possible way.” (46), and further supported by our finding that IPI tuning of pC1 neurons is correlated with behavioral responses. We have now added a brief summary of Riabinina et al. in the results.

*7) In the first sentence of the fourth paragraph of the Introduction, the word “perceive” here is incorrect. “Receive” is more appropriate here. In other places in the manuscript the authors correctly use the word “perceive” in the context of decoding the sensory signals*.

We have now corrected this word.

*8) In the fourth paragraph of the Introduction, upon mention of the projections to AMMC, the authors should mention that the auditory JO neurons also have a direct input to the Giant Fiber. Relevant citations are Lehnert et al., which they already cite elsewhere in a different context, and*
[43]*, J Neuroscience*.

We have now mentioned the connections of JON to the Giant Fiber neurons and added the relevant citations.